# FLATQUANT: Flatness Matters for LLM Quantization

Yuxuan Sun [*1]  Ruikang Liu [*2]  Haoli Bai [†1]  Han Bao [1]  Kang Zhao [1]  Yuening Li [3]
Jiaxin Hu [1]  Xianzhi Yu [1]  Lu Hou [1]  Chun Yuan [1]  Xin Jiang [1]  Wulong Liu [1]  Jun Yao [1]

## Abstract

Recently, quantization has been widely used for the compression and acceleration of large language models (LLMs). Due to the outliers in LLMs, it is crucial to flatten weights and activations to minimize quantization error with equally spaced quantization points. Prior research explores various pre-quantization transformations to suppress outliers, such as per-channel scaling and Hadamard transformation. However, we observe that these transformed weights and activations can still exhibit steep and dispersed distributions. In this paper, we propose FLATQUANT (**F**ast and **L**earnable **A**ffine **T**ransformation), a new post-training quantization approach that enhances the flatness of weights and activations. Our approach identifies optimal affine transformations for each linear layer, calibrated in hours via a lightweight objective. To reduce runtime overhead of affine transformation, we apply Kronecker product with two lightweight matrices, and fuse all operations in FLATQUANT into a single kernel. Extensive experiments demonstrate that FLATQUANT establishes a new state-of-the-art benchmark for quantization. For example, it achieves less than **1**% accuracy drop for W4A4 quantization on the LLaMA-3-70B model, surpassing SpinQuant by **7.5**%. Additionally, it provides up to **2.3x** prefill speedup and **1.7x** decoding speedup compared to the FP16 model. Code is available at: https://github.com/ruikangliu/FlatQuant.

## 1. Introduction

Recent large language models (LLMs) have achieved remarkable success across a wide range of tasks with an in-

creasing number of parameters (Achiam et al., 2023; Jiang et al., 2023; Yang et al., 2024; Dubey et al., 2024). However, the growth of model size also incurs a significant increase in computation and memory overhead. As a result, reducing the computational and memory demands of LLMs has emerged as a critical research direction, and quantization is one of the most effective solutions (Frantar et al., 2022; Lin et al., 2023; Dettmers et al., 2022; Xiao et al., 2023).

Quantization decreases the memory footprint and accelerates the inference, by reducing the precision of model parameters and activations. Quantization error is a commonly used metric to measure the performance of quantization methods (Nagel et al., 2020; Bai et al., 2020; Li et al., 2021). One key factor that affects the quantization error is the ***flatness*** of weights and activations. Intuitively, when the distribution of weights and activations is sharp and there exist multiple outspread values, quantizing them to the same quantized value usually incurs a large quantization error (Chmiel et al., 2020; Li et al., 2024). Moreover, as LLMs generate outputs layer by layer, a reduced quantization error also flattens the error landscape propagated across Transformer layers.

Nevertheless, it is non-trivial to get a flat distribution of weights and activations in LLMs. LLMs are known to have extreme outliers over activations (Dettmers et al., 2022; Xiao et al., 2023) and pivot tokens (Liu et al., 2024a; Sun et al., 2024). To alleviate this problem, various pre-quantization transformations are proposed to mitigate the impact of outliers (Xiao et al., 2023; Ashkboos et al., 2024; Liu et al., 2024c; Ma et al., 2024). However, we revisit these transformations and find them still sub-optimal in promoting flatness. For instance, per-channel scaling (Xiao et al., 2023; Shao et al., 2023) aims to balance the outliers between weights and activations, but it falls short of distributing outliers over the non-outlier channels. The recent Hadamard transformation (Ashkboos et al., 2024; Lin et al., 2024) attempts to solve this problem, while the individual characteristics of each linear layer are not considered. Moreover, the linear transformation introduced by these methods inevitably introduces extra inference overhead that affects the overall speedup of quantization.

In this work, we introduce FLATQUANT, a new post-training quantization approach. The name has dual significance: it

---

[*]Equal contribution [1]Huawei Noah's Ark Lab [2]Shenzhen International Graduate School, Tsinghua University [3]The Chinese University of Hong Kong. Correspondence to: Haoli Bai <baihaoli@huawei.com>.

*Proceedings of the 42nd International Conference on Machine Learning*, Vancouver, Canada. PMLR 267, 2025. Copyright 2025 by the author(s).

stands for the proposed method (**F**ast and **L**earnable **A**ffine **T**ransformation) and emphasizes its goal of achieving flatter distributions of weights and activations, which are crucial for effective quantization. FLATQUANT aims to identify the optimal affine transformation for each linear layer, employing a lightweight, block-wise training strategy over the calibration data. To minimize the inference overhead associated with affine transformations, FLATQUANT harnesses the efficiency of Kronecker product with two lightweight matrices, reducing both the memory and computational demands. Our approach is compatible with various quantization techniques such as learnable clipping, and can be applied to various quantization settings, e.g., weight-only quantization or KV cache quantization. Additionally, as the affine transformations in FLATQUANT are memory bound, we further fuse them together with quantization into a single kernel, minimizing the global memory access and kernel lunch overhead. Lastly, extensive experiments over various tasks (e.g., language modeling and question answering) and LLM families (e.g., LLaMA and Qwen) are conducted to assess FLATQUANT. Our empirical results demonstrate that FLATQUANT establishes new state-of-the-art quantization results w.r.t. both accuracy and inference latency.

The contributions of this work are summarized below:

- We highlight the significance of achieving flatness for LLM quantization, demonstrating that flat distributions of weights and activations facilitate quantization and reduce error propagation across Transformer layers.

- We introduce FLATQUANT, a new post-training quantization method with fast and learnable affine transformations optimized for each linear layer. The approach is empirically demonstrated to enhance the flatness of both weights and activations in LLMs.

- Extensive experiments demonstrate that FLATQUANT sets new state-of-the-art results for quantization. To the best of our knowledge, we are the first to achieve $\leq 1\%$ accuracy drop with simply round-to-nearest W4A4 quantization on the LLaMA-3-70B model.

- We also design an efficient kernel that fuses both affine transformation and quantization, leading to **2.3x** prefill speedup and **1.7x** decoding speedup under W4A4 quantization when compared to the FP16 baseline.

## 2. Motivation

### 2.1. Preliminaries on LLM Quantization

The inference of LLM typically has two stages: 1) the prefill stage, which creates a key-value cache (KV Cache) layer by layer from the input sequence; and 2) the decoding stage, where the model autoregressively generates tokens based on previous KV cache. Quantization is a common practice to reduce the model size and inference latency. It converts the full-precision weights $\mathbf{W} \in \mathbb{R}^{m \times n}$ or activations $\mathbf{X} \in \mathbb{R}^{k \times n}$ of linear layers (i.e., $\mathbf{Y} = \mathbf{X}\mathbf{W}^\top$), and optionally the KV cache to low-bit representations. For instance, $b$-bit weight quantization can be represented as follows:

$$\hat{\mathbf{W}} = \mathcal{Q}_b(\mathbf{W}) = s \cdot \Pi_{\Omega(b)}(\mathbf{W}/s), \qquad (1)$$

where $s$ is the quantization step size, $\Pi(\cdot)$ is the projection function and $\Omega(b) = \{0, 1, ..., 2^b - 1\}$ is the set of $b$-bit integer points. For simplicity of notation, we denote $\mathcal{Q}(\cdot)$ as the general quantization function in the rest of this paper.

As recent works suggest (Xiao et al., 2023; Shao et al., 2023; Xi et al., 2023), LLMs exhibit persistent outliers in activations, posing significant challenges for quantization. Various works are proposed to suppress outliers to improve the quantized LLMs. Two commonly used methods are per-channel scaling (Xiao et al., 2023; Lin et al., 2023; Wei et al., 2023) and Hadamard transformation or its variants (Xi et al., 2023; Ashkboos et al., 2024; Lin et al., 2024).

**Per-channel Scaling.** The input activations $\mathbf{X}$ of LLMs are often rich in outliers. To mitigate their impact on quantization, a popular way is to apply channel-wise scaling over weights and activations (Xiao et al., 2023), i.e., $\mathbf{Y} = (\mathbf{X}\text{diag}(\boldsymbol{c})^{-1}) \cdot (\text{diag}(\boldsymbol{c})\mathbf{W}^\top)$, where $\boldsymbol{c} \in \mathbb{R}^n$ is the channel-wise scaling factor. The scaling vector smooths the activations by jointly considering the magnitudes of input activations and weights, i.e. $\boldsymbol{c}_j = \max(|\mathbf{X}_j|^\alpha)/\max(|\mathbf{W}_j|^{1-\alpha})$. The scaled weights $\text{diag}(\boldsymbol{c})\mathbf{W}^\top$ can be merged to eliminate the runtime computation. Additionally, Wei et al. (2023) introduces channel-wise shifting, i.e., $(\mathbf{X} - \boldsymbol{z})\text{diag}(\boldsymbol{c})^{-1}$, to further mitigate the impact of outliers, and Shao et al. (2023) treats both $\text{diag}(\boldsymbol{c})$ and $\boldsymbol{z}$ as learnable parameters.

**Hadamard Transformation.** Recent works find Hadamard matrices $\mathbf{H} \in \{+1, -1\}^{n \times n}$ are particularly helpful in smoothing out outliers in activations (Xi et al., 2023; Ashkboos et al., 2024; Lin et al., 2024). In contrast to per-channel scaling which only adjusts the diagonal elements in the view of matrix multiplication, Hadamard transformation rotates the channels of both activations and weights, re-distributing the outliers among all channels to effectively eliminate them. Thanks to the orthogonality of Hadamard matrices (i.e., $\mathbf{H}^\top\mathbf{H} = \mathbf{I}$), the following equivalency holds: $\mathbf{Y} = \mathbf{X}\mathbf{W}^\top = (\mathbf{X}\mathbf{H})(\mathbf{H}^\top\mathbf{W}^\top)$. The transformed weight $\mathbf{W}\mathbf{H}$ can be similarly pre-processed offline to reduce additional runtime overhead.

### 2.2. The Flatness for Quantization

**The Flatness of Weights and Activations.** Flat tensors are intuitively easier to quantize after removing outliers, a.k.a tensors with low kurtosis (Chmiel et al., 2020; Li et al.,

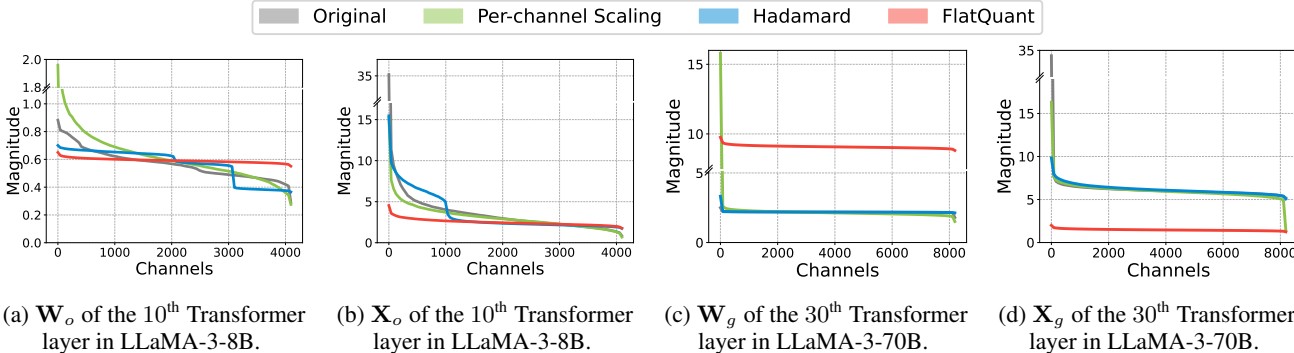

(a) $\mathbf{W}_o$ of the 10th Transformer layer in LLaMA-3-8B.

(b) $\mathbf{X}_o$ of the 10th Transformer layer in LLaMA-3-8B.

(c) $\mathbf{W}_g$ of the 30th Transformer layer in LLaMA-3-70B.

(d) $\mathbf{X}_g$ of the 30th Transformer layer in LLaMA-3-70B.

Figure 1: Distributions of weights and inputs from LLaMA-3-8B and LLaMA-3-70B, sorted by the channel magnitudes (i.e., the Frobenius norm) in descending order. In a Transformer layer, $\mathbf{W}_o$ and $\mathbf{X}_o$ denote the weight matrix and input of the output projection layer in the self-attention layer, respectively. $\mathbf{W}_g$ and $\mathbf{X}_g$ denote the weight and input of the gated linear layer of the feed-forward network, respectively. More visualizations can be found in Appendix D.

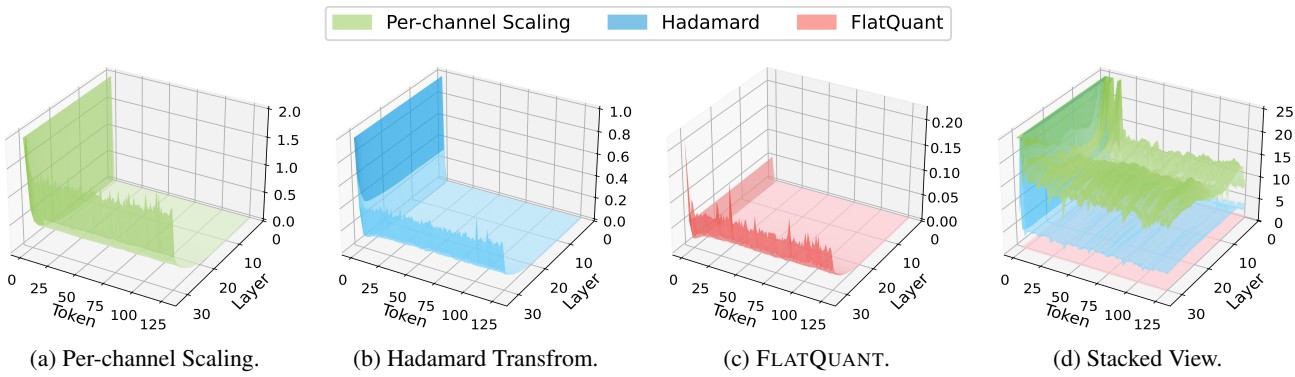

(a) Per-channel Scaling.

(b) Hadamard Transfrom.

(c) FLATQUANT.

(d) Stacked View.

Figure 2: The mean squared error (MSE) of quantization across Transformer layers and input sequence in LLaMA-3-8B. Figure 2a-2c plot the MSE surface of each method, while Figure 2d overlays these surfaces by dividing each MSE with that of FLATQUANT. More details and visualizations can be found in Appendix D.

2024). Figure 1 displays the distributions of both the original and transformed weights and activations, sorted by the channel magnitudes in descending order. The flat weights and activations with horizontal envelopes are usually preferred by quantization. Compared with the original distributions, pre-quantization transformations can yield flatter activations (e.g., Figure 1b, 1d) but still with their limitations. Per-channel scaling flattens activations at the cost of steeper weight envelops (e.g., Figure 1a, 1c). While Hadamard transformation produces better flatness for both activations and weights than per-channel scaling, it still sometimes generates unsatisfactory distributions (e.g., Figure 1a, 1b). In contrast, FLATQUANT, as will be elaborated in Section 3, consistently flattens both weights and activations.

**The Flatness of Quantization Error Landscape.** The quantization error inevitably propagates, and it is insightful to show how pre-quantization transformations mitigate this issue. We plot the two-dimensional landscape of mean squared error (MSE) in Figure 2. First, it is observed that massive quantization errors occur at initial tokens, a.k.a.

pivot tokens (Liu et al., 2024a), which contain massive outliers (Sun et al., 2024). Both per-channel scaling and Hadamard transformation are powerless to such errors (i.e., Figure 2a-2b). Instead, FLATQUANT shows much lower error at these pivot tokens from Figure 2c. Second, the quantization error increases layer-wisely, but is less evident along the input sequence. According to Figure 2d, FLATQUANT is the best in controlling the error propagation, followed by Hadamard transformation and lastly the per-channel scaling.

## 3. Method

We now introduce FLATQUANT with an overview in Figure 3. The proposed method has dual significance: it employs fast and learnable affine transformations, which also produce flat weights and activations for easy quantization.

### 3.1. Fast and Learnable Affine Transformation

We first introduce FLATQUANT for a standard linear layer, and will discuss its integration with the Transformer archi-

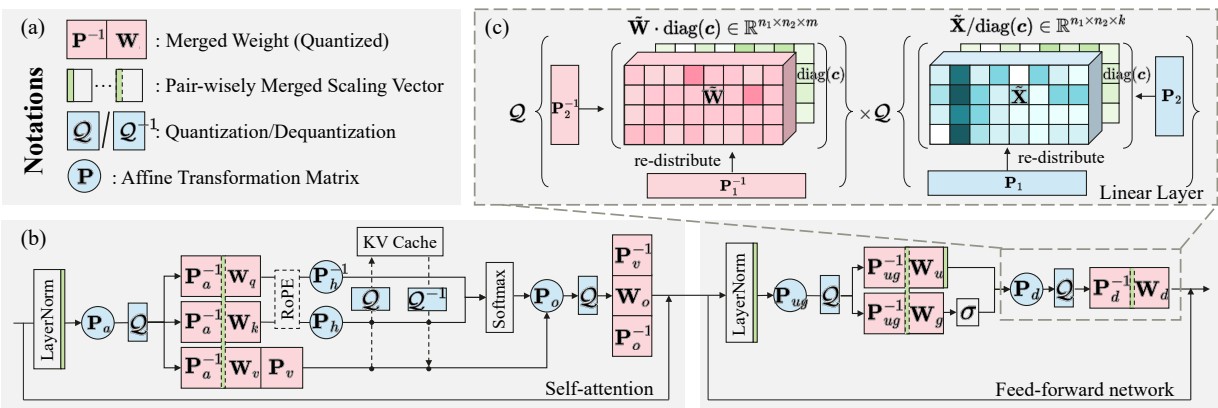

Figure 3: The overall framework of FLATQUANT. (a): necessary notations of FLATQUANT; (b): the integration of FLATQUANT with a conventional LLaMA layer, where merged parameters are grouped in red, online transformation and quantization functions in blue, and merged scaling vectors in green; (c): the exemplary view of FLATQUANT applied for the down-projection layer, where the scaling vector $\mathrm{diag}(\boldsymbol{c})$ over $\tilde{\mathbf{X}}$ is merged to $\mathbf{W}_u$ in practice.

tecture in Section 3.2. A primary objective of FLATQUANT is to find the best affine transformation for each linear layer to quantize. Ideally, given $\mathbf{Y} = \mathbf{XW}^\top$, one can identify the optimal invertible matrix $\mathbf{P}^* \in \mathbb{R}^{n \times n}$ by

$$\mathbf{P}^* = \arg\min_{\mathbf{P}} \|\mathbf{Y} - \mathcal{Q}(\mathbf{XP})\mathcal{Q}(\mathbf{P}^{-1}\mathbf{W}^\top)\|_F^2, \quad (2)$$

as studied in (Ma et al., 2024). The weights $\mathbf{P}^{-1}\mathbf{W}^\top$ can be pre-computed offline akin to (Ashkboos et al., 2024). However, unlike Hadamard matrices that can be reused for all layers, it is computationally expensive to maintain individual $\mathbf{P}$ matrices for different linear layers. In the forward pass, this approach doubles the computational cost and memory access for matrix multiplication. Additionally, it nearly doubles the model storage requirements.

**Kronecker Product.** The key of FLATQUANT is to use Kronecker product of two lightweight matrices as an efficient substitution of the large affine transformation matrix $\mathbf{P} \in \mathbb{R}^{n \times n}$. Specifically, we propose to construct two lightweight matrices, i.e. $\mathbf{P} = \mathbf{P}_1 \otimes \mathbf{P}_2$, where $\mathbf{P}_1 \in \mathbb{R}^{n_1 \times n_1}, \mathbf{P}_2 \in \mathbb{R}^{n_2 \times n_2}$ are invertible matrices in smaller sizes, and $n = n_1 n_2$. Recall the vectorization trick of the Kronecker product, i.e., $\mathrm{vec}(\mathbf{V})(\mathbf{P}_1 \otimes \mathbf{P}_2) = \mathrm{vec}(\mathbf{P}_1^\top \mathbf{V} \mathbf{P}_2)$ for some $\mathbf{V} \in \mathbb{R}^{n_1 \times n_2}$, the matrix multiplication in Equation 2 can be re-written as

$$\mathcal{Q}(\mathbf{XP})\mathcal{Q}(\mathbf{P}^{-1}\mathbf{W}^\top) = \quad (3)$$
$$\mathcal{Q}(\mathbf{P}_1^\top \times_1 \tilde{\mathbf{X}} \times_2 \mathbf{P}_2) \times \mathcal{Q}(\mathbf{P}_1^{-1} \times_1 \tilde{\mathbf{W}} \times_2 (\mathbf{P}_2^{-1})^\top)^\top,$$

where $\tilde{\mathbf{X}} \in \mathbb{R}^{k \times n_1 \times n_2}$ and $\tilde{\mathbf{W}} \in \mathbb{R}^{m \times n_1 \times n_2}$ are reshaped from $\mathbf{X}$ and $\mathbf{W}$ accordingly, and $\times_i$ denotes the reduction over the $i$-th axis. Note that both weights and activations are converted back to matrix before multiplication. This design can save the memory up to $n/2$ times, given that $\frac{n^2}{n_1^2 + n_2^2} \leq$

$\frac{n^2}{2n_1 n_2} = \frac{n}{2}$, with the equality holds when $n_1 = n_2 = \sqrt{n}$. Moreover, the computation saving is $\sqrt{n}/2$ times with the same optimal condition. In practice, we select $n_1^*, n_2^* = \arg\min(n_1 + n_2)$, s.t. $n_1 n_2 = n$ and $n_1 \leq n_2$. For instance, the optimal configuration is $(n_1^*, n_2^*) = (64, 128)$ for $n = 8192$. We find such a strategy gives the best speedup without compromising performance, as shown in Figure 5. The affine transformations are therein pretty light-weight, e.g., they only take 2.61% of the overall computational FLOPs and 3.41MB extra memory for LLaMA-2-7B. More details of inference overhead are in Appendix B.2.

**Per-channel Scaling.** To enhance the ability to balance outliers between the weights and activations, FLATQUANT explicitly introduces a learnable scaling vector $\mathrm{diag}(\boldsymbol{c}) \in \mathbb{R}^n$ prior to the pre-quantization transformation, as illustrated in Figure 3 (c). Following (Xiao et al., 2023), the scaling vector can be merged pair-wisely to the preceding layer normalization or linear layers, thereby incurring no additional inference overhead.

**Learnable Clipping Thresholds.** To further reduce the potential outlier after the above transformation, we combine learnable clipping thresholds $\alpha_w, \alpha_a \in (0, 1)$ after sigmoid functions on both weight and activation for each linear layer, together with the KV cache. While previous studies (Jacob et al., 2018; Frantar et al., 2022; Ashkboos et al., 2024) demonstrate that grid search is valid to find reasonable clipping thresholds, we observe that learning the clipping thresholds yields better results. These parameters are layer-specific and can be jointly optimized with the linear transformation matrices $\mathbf{P}$ and scaling vector $\mathrm{diag}(\boldsymbol{c})$.

**The Training Objective.** We follow post-training quantization and sequentially minimize the mean squared error (MSE) by quantization over a small amount of calibra-

tion data (e.g., 128 randomly sampled sentences) for each Transformer block. The training objective for the $l$-th Transformer block is

$$\min_{\Theta}\left\|\mathcal{F}_l(\mathbf{X}) - \hat{\mathcal{F}}_l(\mathbf{X};\Theta)\right\|_F^2, \qquad (4)$$

where $\mathcal{F}_l(\cdot)$ and $\hat{\mathcal{F}}_l(\cdot)$ denote the original and the quantized Transformer block, $\Theta = \{\mathbf{P}, \boldsymbol{c}, \alpha_a, \alpha_w\}$ is abbreviated for all learnable parameters within the block. The transformation matrices within a Transformer block will be explained in Section 3.2. To compute the matrix inversion in Equation 3 efficiently and accurately, we adopt the singular value decomposition together with automatic mixed precision. See Appendix B.1 for details. We also tried with training multiple Transformer blocks together but found similar performance at higher training costs. Finally, the training progress with Equation 4 leads to flat weights and activations, as will be shown in Section 4.4.

## 3.2. Integration with the Transformer Architecture

We now illustrate the integration of FLATQUANT with a Transformer block based on an LLaMA-like architecture. Following the conventional practices, we employ low-bit matrix multiplications for all linear layers, while keeping layer normalization layers, pre-quantization transformations, RoPE embeddings, and attention scores in FP16.

**Self-Attention.** The self-attention module is equipped with four transformations $\{\mathbf{P}_a, \mathbf{P}_o, \mathbf{P}_h, \mathbf{P}_v\}$. Specifically, $\mathbf{P}_a$ is applied to flatten the input activation for the query, key, and value projections, while $\mathbf{P}_o$ smooths the input activation for the output projection. $\mathbf{P}_h$ and $\mathbf{P}_v$ are used to transform the key and value cache head by head, respectively. Note that we only decompose $\mathbf{P}_a$ and $\mathbf{P}_o$, but leave $\mathbf{P}_h$ and $\mathbf{P}_v$ in their original shape. This is because per-head quantization already facilitates cheap transformations, given that the head size is significantly smaller than the full hidden size. Moreover, we further fuse $\mathbf{P}_o$ with $\mathbf{P}_v$ to reduce overhead, as inspired by QuaRot (Ashkboos et al., 2024). Our empirical results show this fusion does not result in additional loss of accuracy.

**Feed-forward Network.** The feed-forward network (FFN) employs two transformation matrices, i.e., $\mathbf{P}_{ug}$ and $\mathbf{P}_d$. $\mathbf{P}_{ug}$ is applied to flatten the input of the feed-forward network after layer normalization, while $\mathbf{P}_d$ flattens the input for the down projection layer. Both transformations are decomposed to minimize the inference overhead. Additionally, the per-channel scaling of $\mathbf{P}_d$ is merged into the weight of up projection layer, ensuring no additional computational overhead.

**Layer Normalization.** Recall that QuaRot (Ashkboos et al., 2024) and SpinQuant (Liu et al., 2024c) modify the LayerNorm to RMSNorm and merge orthogonal transformations into preceding layers for efficiency. Nonetheless, the

residual connection of the "pre-norm" architecture would constrain all Transformer blocks to share the same transformation after RMSNorm. Instead, FLATQUANT preserves the LayerNorm, and allows the use of fast and learnable affine transformations in Section 3.1 after LayerNorm for different layers, thereby enhancing the expressiveness.

## 3.3. Efficient Kernel Design

We design an efficient kernel for FLATQUANT that integrates both affine transformations and quantization into a single operation. This design is motivated by two key factors. First, $\mathbf{P}_1^\top \times_1 \tilde{\mathbf{X}} \times_2 \mathbf{P}_2$ exhibits low computational intensity with Kronecker product of two lightweight matrices, making both prefilling and decoding predominantly memory-bound. Second, the quantization is also known to be memory-bound.

To address these issues, we fuse $\mathcal{Q}(\mathbf{P}_1^\top \times_1 \tilde{\mathbf{X}} \times_2 \mathbf{P}_2)$ into a single kernel using OpenAI Triton (Tillet et al., 2019). Specifically, we load the entire $\mathbf{P}_1 \in \mathbb{R}^{n_1 \times n_1}$ and $\mathbf{P}_2 \in \mathbb{R}^{n_2 \times n_2}$ into SRAM. Each thread block slices a tiling block $\bar{\mathbf{X}} \in \mathbb{R}^{n_1 \times n_2}$ from $\tilde{\mathbf{X}}$, performs the matrix multiplication $\mathbf{P}_1\bar{\mathbf{X}}\mathbf{P}_2$, and quantizes the results on the fly. Throughout this process, all intermediate results are stored in SRAM before finally being written back to the global memory. This design thereby eliminates redundant memory accesses of intermediate results and reduces the kernel launch overhead. Finally, given the output above, we follow QuaRot (Ashkboos et al., 2024) to adopt the CUTLASS kernel for INT4 matrix multiplication, and FlashInfer (Ye, 2023) for KV cache quantization. Further details of the kernel design are provided in the Appendix B.3.

## 4. Experiments

### 4.1. Settings

**Evaluation and Baselines.** We primarily evaluate FLATQUANT on the series of LLaMA-2 (Touvron et al., 2023) and LLaMA-3 (Dubey et al., 2024) models, and results on Qwen and DeepSeek families can be found in Appendix C.1. Following previous works (Shao et al., 2023; Ashkboos et al., 2024), we report the perplexity (PPL) of language generation tasks on the WikiText2 (Merity et al., 2016) and C4 (Raffel et al., 2020) datasets. For commonsense reasoning tasks, we use six zero-shot evaluation tasks, including ARC-Challenge, ARC-Easy (Clark et al., 2018), HellaSwag (Zellers et al., 2019), LAMBADA (Paperno et al., 2016), PIQA (Bisk et al., 2020), and WinoGrande (Sakaguchi et al., 2021). We compare FLATQUANT against popular INT4 post-training quantization methods, including SmoothQuant (Xiao et al., 2023), OmniQuant (Shao et al., 2023), AffineQuant (Ma et al., 2024), QUIK-4B (Ashkboos et al., 2023), and two recent

| Method | W Quantizer | WikiText-2 | | | | | C4 | | | | |
|---|---|---|---|---|---|---|---|---|---|---|---|
| | | 2-7B | 2-13B | 2-70B | 3-8B | 3-70B | 2-7B | 2-13B | 2-70B | 3-8B | 3-70B |
| FP16 | - | 5.47 | 4.88 | 3.32 | 6.14 | 2.86 | 7.26 | 6.73 | 5.71 | 9.45 | 7.17 |
| SmoothQuant | RTN | 83.12 | 35.88 | 26.01 | 210.19 | 9.60 | 77.27 | 43.19 | 34.61 | 187.93 | 16.90 |
| OmniQuant | RTN | 14.74 | 12.28 | - | - | - | 21.40 | 16.24 | - | - | - |
| AffineQuant | RTN | 12.69 | 11.45 | - | - | - | 15.76 | 13.97 | - | - | - |
| QuaRot | RTN | 8.56 | 6.10 | 4.14 | 10.60 | 55.44 | 11.86 | 8.67 | 6.42 | 17.19 | 79.48 |
| SpinQuant | RTN | 6.14 | 5.44 | 3.82 | 7.96 | 7.58 | 9.19 | 8.11 | 6.26 | 13.45 | 15.39 |
| FLATQUANT | RTN | **5.79** | **5.12** | **3.55** | **6.98** | **3.78** | **7.79** | **7.09** | **5.91** | **11.13** | **7.86** |
| QUIK-4B | GPTQ | 8.87 | 7.78 | 6.91 | - | - | - | - | - | - | - |
| QuaRot | GPTQ | 6.10 | 5.40 | 3.79 | 8.16 | 6.60 | 8.32 | 7.54 | 6.12 | 13.38 | 12.87 |
| SpinQuant | GPTQ | 5.96 | 5.24 | 3.70 | 7.39 | 6.21 | 8.28 | 7.48 | 6.07 | 12.19 | 12.82 |
| FLATQUANT | GPTQ | **5.78** | **5.11** | **3.54** | **6.90** | **3.77** | **7.86** | **7.11** | **5.92** | **11.21** | **7.93** |

Table 1: WikiText-2 and C4 perplexity of 4-bit weight & acitvation quantized LLaMA models.

state-of-the-art methods QuaRot (Ashkboos et al., 2024) and SpinQuant (Liu et al., 2024c).

**Implementation Details.** We implement FLATQUANT based on Huggingface (Wolf, 2019) and PyTorch (Paszke et al., 2019). We adopt the AdamW optimizer with an initial learning rate of 5e-3 and employ a cosine annealing learning rate decay schedule. The learning rate for clipping thresholds is 5e-2. FLATQUANT is trained for 15 epochs on a calibration set comprising 128 sentences from WikiText-2, each sampled with 2048 tokens. The batch size is set to 4. The default calibration procedure costs approximately 26GB of GPU memory and about 0.9 hours for LLaMA-3-8B on a single GPU. FLATQUANT is robust to initialization, and we employ random affine transformation matrices as the starting point. Further details about implementation and calibration time are provided in Appendix B.1.

**Quantization.** We adopt per-channel and per-token symmetric quantization for weights and activations, respectively. For fair comparisons with QuaRot and SpinQuant, we employ both round-to-nearest (RTN) and GPTQ as the weight quantizer, where GPTQ uses the same calibration data for both closed-form weight updates and training. Nonetheless, our empirical results suggest that FLATQUANT with RTN is sufficient to be competitive. FLATQUANT can be also used for KV cache quantization, where group-wise asymmetric quantization with the size of 128 is applied. This matches the head dimension of LLaMA, as suggested in previous studies (Zhao et al., 2024; Ashkboos et al., 2024), to leverage the memory-bound characteristics of self-attention.

### 4.2. Main Results

**Results on Language Generation Tasks.** Table 1 presents the PPL results for FLATQUANT with and without the GPTQ weight quantizer on the WikiText-2 and C4 datasets. As can be seen, FLATQUANT with RTN weight

quantizer consistently outperforms previous SOTA quantization methods across all major benchmarks. For the LLaMA-2-70B model, FLATQUANT achieves a PPL score just 0.23 higher than the FP16 baseline, underscoring the effectiveness of our approach. For LLaMA-3-8B, FLATQUANT reduces the PPL from 7.39 (SpinQuant) to 6.98, narrowing the gap with the FP16 baseline to 0.84. Notably, FLATQUANT with RTN exhibits performance comparable to those with GPTQ but takes significantly less calibration time. This is particularly helpful in reducing the time consumption to deploy FLATQUANT in practice. These results highlight the efficacy of our proposed learnable transformations in enhancing flatness and mitigating the impact of outliers in both weights and activations, thereby establishing a new SOTA in low-bit LLM quantization.

**Results on Zero-shot QA Tasks.** We extend our evaluation to six zero-shot commonsense QA tasks, as shown in Table 2. For a fair comparison, we reproduce QuaRot [1] and SpinQuant [2] with their official implementations and released checkpoints, evaluating all methods with the same version of lm-eval-harness framework (Gao et al., 2021). As can be seen, FLATQUANT significantly narrows the performance gap between quantized models and the FP16 baseline. Specifically, while the LLaMA-3 models are shown to be challenging for quantization (Huang et al., 2024), FLATQUANT perform well with an accuracy loss of 2.00% for LLaMA-3-8B and 0.94% for LLaMA-3-70B. Notably, while QuaRot with RTN largely lags behind QuaRot with GPTQ by an average accuracy gap over 4%, FLATQUANT with RTN can already obtain comparable results to GPTQ.

Due to limited space, we leave more experimental results in Appendix C, such as exploration to other quantization set-

---

[1] https://github.com/spcl/QuaRot
[2] https://github.com/facebookresearch/SpinQuant

| Model | Method | W Quantizer | ARC-C | ARC-E | HellaSwag | LAMBADA | PIQA | Winogrande | Avg |
|-------|--------|-------------|-------|-------|-----------|---------|------|------------|-----|
| **2-7B** | FP16 | - | 46.16 | 74.54 | 75.98 | 73.92 | 79.05 | 69.06 | 69.79 |
| | QuaRot | RTN | 36.60 | 61.41 | 65.07 | 48.06 | 72.20 | 63.06 | 57.73 |
| | SpinQuant | RTN | 39.42 | 65.32 | 71.45 | 66.16 | 75.30 | 63.46 | 63.52 |
| | FLATQUANT | RTN | 43.26 | 72.05 | 73.64 | 72.04 | 77.26 | 69.53 | **67.96** |
| | QuaRot | GPTQ | 42.32 | 68.35 | 72.53 | 65.40 | 76.33 | 65.11 | 65.01 |
| | SpinQuant | GPTQ | 41.72 | 69.28 | 72.90 | 71.28 | 76.17 | 66.06 | 66.23 |
| | FLATQUANT | GPTQ | 43.00 | 71.21 | 73.31 | 72.06 | 77.53 | 67.72 | **67.47** |
| **2-13B** | FP16 | - | 49.15 | 77.44 | 79.39 | 76.73 | 80.47 | 72.14 | 72.55 |
| | QuaRot | RTN | 42.83 | 69.95 | 73.54 | 65.62 | 77.69 | 67.88 | 66.25 |
| | SpinQuant | RTN | 43.69 | 72.43 | 75.52 | 72.42 | 78.40 | 68.90 | 68.56 |
| | FLATQUANT | RTN | 48.04 | 76.64 | 77.59 | 76.60 | 79.38 | 70.24 | **71.42** |
| | QuaRot | GPTQ | 45.48 | 73.27 | 76.03 | 69.01 | 79.05 | 70.64 | 68.91 |
| | SpinQuant | GPTQ | 49.15 | 77.19 | 76.86 | 73.86 | 78.67 | 69.85 | 70.93 |
| | FLATQUANT | GPTQ | 48.38 | 76.94 | 77.88 | 76.40 | 79.65 | 70.56 | **71.64** |
| **2-70B** | FP16 | - | 57.17 | 81.02 | 83.81 | 79.60 | 82.70 | 77.98 | 77.05 |
| | QuaRot | RTN | 52.22 | 76.60 | 79.96 | 74.61 | 81.12 | 76.32 | 73.47 |
| | SpinQuant | RTN | 55.03 | 79.17 | 81.76 | 78.87 | 81.45 | 74.27 | 75.09 |
| | FLATQUANT | RTN | 56.14 | 80.30 | 83.01 | 79.60 | 82.75 | 77.90 | **76.62** |
| | QuaRot | GPTQ | 55.46 | 79.76 | 81.58 | 79.35 | 81.83 | 76.09 | 75.68 |
| | SpinQuant | GPTQ | 55.38 | 79.04 | 82.57 | 78.75 | 82.37 | 78.22 | 76.06 |
| | FLATQUANT | GPTQ | 56.40 | 80.09 | 82.91 | 80.01 | 82.92 | 76.87 | **76.53** |
| **3-8B** | FP16 | - | 53.50 | 77.57 | 79.12 | 75.51 | 80.74 | 72.93 | 73.23 |
| | QuaRot | RTN | 38.65 | 66.54 | 68.82 | 57.20 | 71.82 | 65.04 | 61.34 |
| | SpinQuant | RTN | 45.73 | 71.38 | 74.07 | 67.67 | 76.66 | 66.38 | 66.98 |
| | FLATQUANT | RTN | 50.00 | 75.80 | 76.80 | 72.91 | 79.16 | 72.69 | **71.23** |
| | QuaRot | GPTQ | 45.73 | 70.83 | 72.97 | 62.70 | 75.35 | 67.17 | 65.79 |
| | SpinQuant | GPTQ | 47.27 | 74.20 | 74.55 | 70.29 | 77.37 | 68.51 | 68.70 |
| | FLATQUANT | GPTQ | 50.51 | 75.88 | 76.49 | 73.20 | 79.00 | 72.93 | **71.33** |
| **3-70B** | FP16 | - | 64.25 | 85.94 | 84.93 | 79.37 | 84.44 | 80.74 | 79.95 |
| | QuaRot | RTN | 22.18 | 34.30 | 32.15 | 13.35 | 57.67 | 52.49 | 35.36 |
| | SpinQuant | RTN | 44.03 | 69.07 | 74.57 | 63.34 | 76.99 | 65.98 | 65.66 |
| | FLATQUANT | RTN | 62.12 | 84.97 | 83.95 | 78.73 | 84.28 | 80.03 | **79.01** |
| | QuaRot | GPTQ | 49.49 | 74.37 | 77.22 | 71.69 | 78.89 | 71.03 | 70.45 |
| | SpinQuant | GPTQ | 51.96 | 77.40 | 77.29 | 71.90 | 79.33 | 72.06 | 71.66 |
| | FLATQUANT | GPTQ | 61.95 | 84.47 | 83.87 | 77.99 | 83.95 | 79.24 | **78.58** |

Table 2: Zero-shot QA task results of 4-bit weight & activation quantized LLaMA models.

tings in Appendix C.3, results on more LLM architectures in Appendix C.1, and MT-bench evaluations in Appendix C.2.

### 4.3. Inference Latency

All experiments of inference latency below are conducted on the RTX3090 GPU. More details of the overall computational FLOPs, kernel profiling, and speedup gains are available in Appendix C.8.

**End-to-end Speedup.** Figure 4 shows the prefill and decoding speedup of FLATQUANT across different batch sizes, with 2048 and 256 tokens for prefill and decoding, respectively. It can be found that even without kernel fusion,

FLATQUANT acheives comparable speed-up with QuaRot, thanks to the Kronecker product of two lightweight matrices. With kernel fusion, FLATQUANT can achieve up to 2.30x speedup for prefill and 1.76x speedup for decoding under the batch size of 64, which is apparently faster than QuaRot (Ashkboos et al., 2024). Although there is still a minor gap compared to the vanilla INT4 quantization, it significantly enhances accuracy and facilitates the deployment of INT4 LLMs in real-world applications.

**Kronecker Product: Sizes and Perplexities.** In Figure 5, we examine the impact of different decomposed matrix sizes in Equation 3 on model performance and speedup.

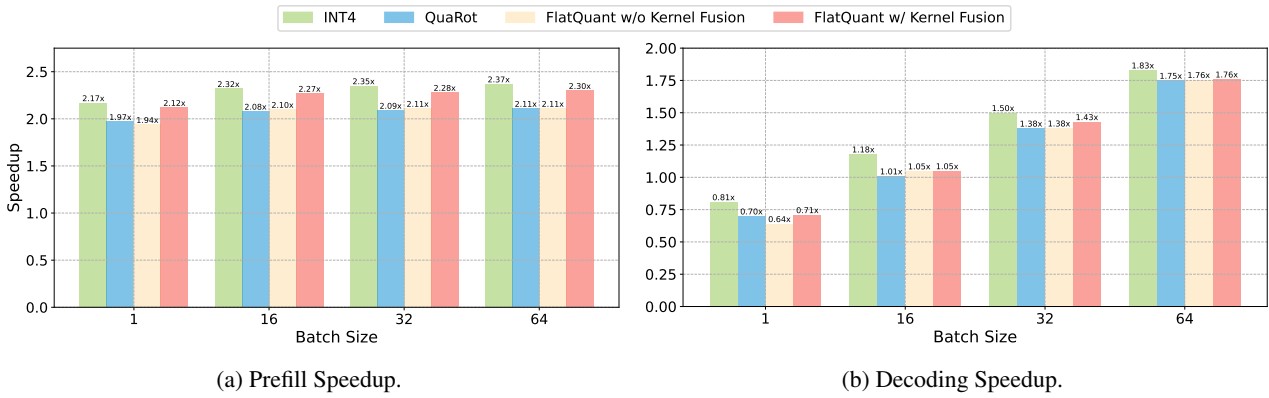

(a) Prefill Speedup.

(b) Decoding Speedup.

Figure 4: Prefill and decoding speedup of LLaMA-2-7B model across different batch sizes. We decode 256 tokens after the prefill on a sequence length of 2048.

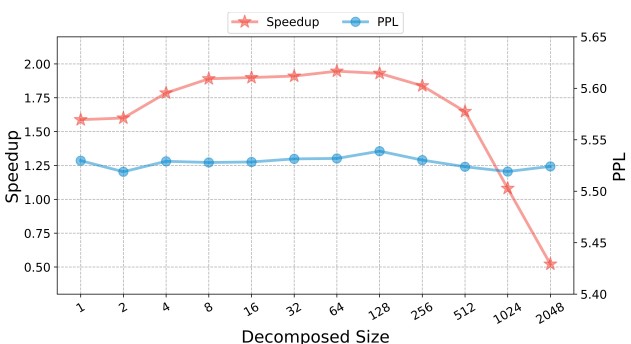

Figure 5: Prefill speedup and WikiText2 PPL results of different decomposed matrix sizes on LLaMA-2-7B model. We decompose the hidden dimension 4096 into $n_1 \times n_2$ and range $n_1$ from 1 to 2048, where $n_1 = 1$ amounts to maintaining a full-size transformation matrix. More details can be found in Appendix C.6.

As shown, the varying sizes of matrices for the Kronecker product significantly affect speedup. However, they have limited impact on the perplexity of generated text. The speedup peaks when $\mathbf{P}_1$ and $\mathbf{P}_2$ are of equal size (i.e., $n_1 = n_2 = \sqrt{n} = 64$), as predicted by our theoretical analysis in Section 3.1. When $n_2$ exceeds 64, the speedup quickly decreases due to irregular memory access patterns for activations. These results further demonstrate FLATQUANT's effectiveness in minimizing inference overhead while maintaining quantization accuracy with the Kronecker product.

**Overhead of Each Online Transformation.** We now investigate the impact of the five online transformations (i.e., $\{\mathbf{P}_a, \mathbf{P}_o, \mathbf{P}_h, \mathbf{P}_{ug}, \mathbf{P}_d\}$) in FLATQUANT on the overall speedup, as shown in Figure 6. Even with five per-layer transformations, FLATQUANT results in a minimal 0.07x end-to-end slowdown, significantly outperforming QuaRot's 0.26x with just three Hadamard transformations. Specifically, FLATQUANT's $\mathbf{P}_d$ causes a 0.04x slowdown due to large FFN intermediate sizes, compared with QuaRot's

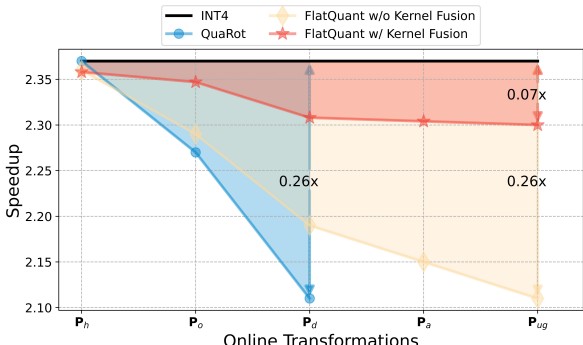

Figure 6: Prefill speedup of LLaMA-2-7B on a sequence length of 2048 under a batch size of 64 after applying different online transformations. We incorporate different online transformations sequentially to gauge their impact on the final speedup. Each point on the x-axis indicates adding a new online transformation.

0.17x. Meanwhile, $\mathbf{P}_o$ results in a 0.01x slowdown, versus QuaRot's 0.1x. The rest transformations (i.e., $\mathbf{P}_a$ and $\mathbf{P}_{ug}$) have an insignificant impact of less than 0.01x. Finally, it can be found that even without kernel fusion, the additional transformations in FLATQUANT is still on par with QuaRot, thanks to the Kronecker product of two lightweight matrices.

### 4.4. Discussions

| LT | PS | LCT | WikiText-2 | C4 | Avg |
|---|---|---|---|---|---|
| | | | 1266.60 | 936.41 | 30.99 |
| ✓ | | | 8.50 | 13.51 | 66.82 |
| ✓ | ✓ | | 7.95 | 12.74 | 67.08 |
| ✓ | | ✓ | 7.11 | 11.47 | 70.72 |
| ✓ | ✓ | ✓ | 6.98 | 11.13 | 71.23 |

Table 3: Ablation study of FLATQUANT's main components on LLaMA-3-8B.

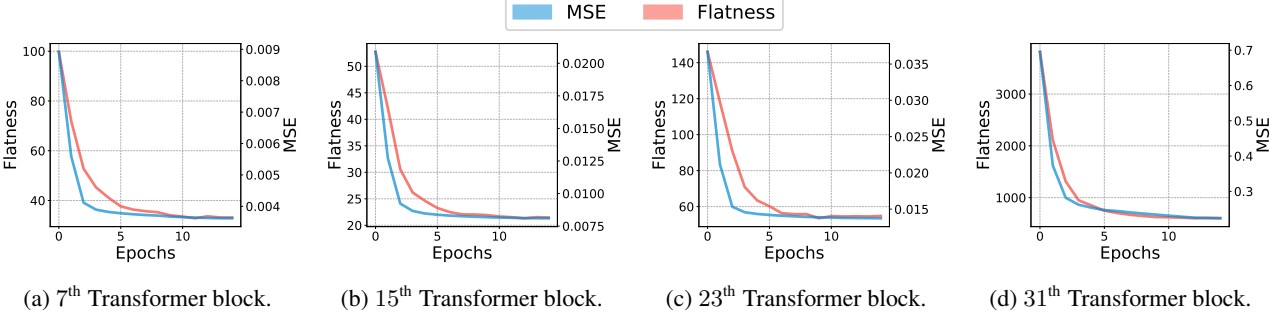

(a) 7ᵗʰ Transformer block.  (b) 15ᵗʰ Transformer block.  (c) 23ᵗʰ Transformer block.  (d) 31ᵗʰ Transformer block.

Figure 7: Flatness and mean squared quantization error (MSE) of different Transformer blocks in LLaMA-3-8B during FLATQUANT's training process. The metric of flatness is calculated as the sum of Euclidean distances $\|\mathbf{d} - \mathbf{d}'\|_2$ for all weights and activations within a Transformer block.

**Ablation Study.** We conduct ablation studies for FLATQUANT focusing on its main components: 1) learnable transformation (LT); 2) per-channel scaling (PS); and 3) learnable clipping thresholds (LCT). Starting from RTN as a baseline, we evaluate the impact of each component on perplexity and the average accuracy on zero-shot QA tasks based on LLaMA-3-8B. As shown in Table 3, enabling LT significantly enhances the accuracy of the quantized model, reducing PPL from 1266.60 to 8.50 on WikiText-2. This shows LT is capable of adaptively flattening the distribution of weights and activations. Additionally, incorporating PS and LCT further improves PPL by 0.55 and 0.84, respectively, demonstrating the necessity of each component to enhance the performance. Due to space limitation, we leave the more comprehensive ablation in Appendix C.4.

**FLATQUANT Leads to Flatness.** To further analyze how FLATQUANT promotes flatness, we quantitatively evaluate the flatness of weights and activations by analyzing their channel-wise magnitude distributions. Specifically, each distribution is represented as a one-dimensional vector $\mathbf{d}$, as illustrated in Figure 1. We measure flatness with the mean squared error (MSE) between the observed distribution $\mathbf{d}$ and an idealized perfectly flat distribution $\mathbf{d}'$. The flat distribution $\mathbf{d}'$ is defined such that all channels possess equal magnitudes and the same $\ell_2$ norm as $\mathbf{d}$, i.e., $\mathbf{d}' = \frac{\|\mathbf{d}\|_2}{\sqrt{N}} \cdot \mathbf{1}_N$, where $N$ is the number of channels and $\mathbf{1}_N$ is an $N$-dimensional vector with all entries equal to one. The Euclidean distance $\|\mathbf{d} - \mathbf{d}'\|_2$ thus serves as our flatness metric, where smaller values indicate distributions closer to uniformity across all channels. In Figure 7, we visualize the evolution of flatness and the training objective (Equation 4) across different Transformer blocks of LLaMA-3-8B during training. With the decreasing training loss, the channel distributions become increasingly flat. This indicates that FLATQUANT learns better transformations to obtain a flatter distribution which ultimately contributes to smaller quantization error, i.e., flatness matters for LLM quantization.

Due to space constraints, we provide additional experiments and discussions in Appendix C.5, including the impact of calibration data, the effect of learnable clipping and mixed-precision schemes. We also analyze inference memory consumption in Appendix C.7. Additional visualizations of flatness and quantization error landscapes are provided in Appendix D.1 and D.2, respectively.

## 5. Conclusions

In this study, we revisit the importance of flat weights and activations for effective quantization, and find existing solutions still produce steep outspread values after the pre-quantization transformation. Therefore, we introduce FLATQUANT, a novel post-training quantization method with the purpose of identifying fast and learnable transformations for each linear layer, to promote the flatness of weights and activations. Extensive experiments demonstrate the superiority of FLATQUANT, e.g., with less than **1**% accuracy drop for W4A4 quantization on the LLaMA-3-70B. Our efficient kernel fusion integrates the affine transformation and quantization, bringing up to **2.3x** and **1.7x** speedup over FP16 inference at the prefill and decoding stages, respectively. We hope this work advances the practical application of low-bit quantization for LLMs.

## Impact Statement

This paper presents work aimed at advancing the field of Machine Learning by improving the efficiency of LLMs through enhanced quantization techniques. These improvements have the potential to lower operational costs and energy consumption, enabling broader access to AI, fostering more equitable use of advanced technologies. However, it is important to recognize that quantization does not address inherent societal biases in training data. Careful consideration and responsible deployment of quantized models are necessary to mitigate ethical risks and ensure their fair use.

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

## A. Related Work

**Quantization for Large Language Models.** Quantization is a crucial technique for reducing memory footprint and accelerating inference by employing fewer bits for storage and computation. Unlike pruning (Ma et al., 2023; Sun et al., 2023; Zhang et al., 2024; Chen et al., 2025), quantization does not alter the network architecture and is usually more competitive in performance under the same compression ratio. Different from the previous pretrained language models (Shen et al., 2020; Bai et al., 2020; 2022), LLMs are shown to exhibit outliers in activation and massive outliers in pivot tokens (Wei et al., 2022; Dettmers et al., 2022; Liu et al., 2024a; Sun et al., 2024), which can severely degrade quantization accuracy, especially on on complex reasoning tasks (Liu et al., 2025). To eliminate the negative impact of outliers, pre-quantization transformations have been widely adopted in weight-activation quantization (Xiao et al., 2023; Wei et al., 2023; Shao et al., 2023; Ma et al., 2024; Ashkboos et al., 2024; Liu et al., 2024c) as well as in fully quantized training (Xi et al., 2023). Additionally, several weight-only quantization methods (Lin et al., 2023; Chee et al., 2024; Tseng et al., 2024) incorporate pre-quantization transformations. Searched or learnable clipping thresholds for weights or activations (Lin et al., 2023; Duanmu et al., 2024; Ashkboos et al., 2024; Liu et al., 2024c; Shao et al., 2023) are also explored to eliminate outliers.

**Per-channel Scaling Transformation.** SmoothQuant (Xiao et al., 2023) employs per-channel scaling to shift the challenge of quantization from activations to weights in weight-activation quantization. Building on this, Wei et al. (2023) additionally introduces channel-wise shifting, while OmniQuant (Shao et al., 2023) utilizes a differentiable approach to learn optimal scaling and shifting parameters. However, the scaling-based methods can negatively impact weight quantization and struggle in low-bit settings, such as W4A4 quantization.

**Hadamard and Orthogonal Transformation.** Recent research (Xi et al., 2023; Tseng et al., 2024; Ashkboos et al., 2024) has shown that the Hadamard transformation is effective in eliminating outliers and lowering the quantization error by redistributing outliers across all channels through matrix multiplication. QuaRot (Ashkboos et al., 2024) is the first to apply Hadamard transformation in the LLM W4A4 PTQ setting, while SpinQuant (Liu et al., 2024c) exploits learnable orthogonal matrices with model-level loss to further alleviate outliers.

**Affine Transformation.** Considering that per-channel scaling corresponds to the diagonal elements of the affine transformation matrix, AffineQuant (Ma et al., 2024) proposes learning the equivalent affine transformation. However, their approach focuses on learning full-size diagonally dominant matrices and employs a gradual mask optimization method, which may hinder the full potential of affine transformation in reducing quantization loss. Moreover, due to the formidable overhead associated with full-size matrix multiplication, AffineQuant can only apply affine transformation to a small fraction of linear layers. In contrast, we employ fast and learnable affine transformations without these limitations, leading to substantial accuracy improvements and practical speedup.

**Pre-quantization Transformations in Other Quantization Tasks.** Inspired by SmoothQuant, AWQ (Lin et al., 2023) introduces activation-aware per-channel scaling to reduce quantization errors in weight-only quantization. QUIP (Chee et al., 2024) and its extension, QUIP# (Tseng et al., 2024), leverage random rotation matrices or Hadamard transformations to enhance incoherence in weight-only quantization. In fully quantized training task, (Xi et al., 2023) propose to utilize a block-diagonal transformation consisting of Hadamard matrices to reduce the quantization error.

## B. Implementation Details

### B.1. Matrix Inversion and Training Cost

A critical aspect to implement FLATQUANT is the computation of the inverse affine transformation matrix $\mathbf{P}^{-1}$. As discussed below, we use singular value decomposition (SVD) and automatic mixed precision to train FLATQUANT, enjoying both training stability and efficiency.

**Direct Inversion and FP32 Training.** One straightforward approach is to use the inverse function provided by PyTorch. However, we find that the precision of this inverse function at FP16 is insufficient. Specifically, $\mathbf{P}\mathbf{P}^{-1}$ does not closely approximate $\mathbf{I}$. The off-diagonal elements are on the order of $1 \times 10^{-3}$, which negatively impacts FLATQUANT's performance during the early stages of training. Therefore, a simple solution is to conduct training in FP32 without Automatic Mixed Precision (AMP) to maintain precision. However, this inevitably increases training time and more GPU memory consumption.

**SVD and AMP Training.**   To further reduce resource requirements during calibration, we propose to employ singular value decomposition for the affine transformation. For any real matrix $\mathbf{P}$, we can decompose it as $\mathbf{P} = \mathbf{U}\boldsymbol{\Sigma}\mathbf{V}^\top$, where $\mathbf{U}$ and $\mathbf{V}$ are orthogonal matrices, and $\boldsymbol{\Sigma}$ is a diagonal matrix. This formulation allows us to easily compute $\mathbf{P}^{-1} = \mathbf{V}\boldsymbol{\Sigma}^{-1}\mathbf{U}^\top$, offering a more computationally efficient method for obtaining the inverse. Notably, this approach reduces the off-diagonal elements of $\mathbf{P}\mathbf{P}^{-1}$ to the order of $1 \times 10^{-6}$ at FP16 precision, enabling us to utilize AMP during calibration. With AMP, we can achieve a 50% reduction in training time and memory usage while maintaining nearly lossless accuracy in most cases. For the orthogonal matrices $\mathbf{U}$ and $\mathbf{V}$, we employ the Cayley parameterization provided by PyTorch [3].

**Comparison of the Two Training Recipes.**   We compare the two training recipes in Table 4. As shown, FP32 training requires more than twice the time of AMP training and necessitates 1.28x more GPU memory under the same setting, while the performance remains relatively close. Thus, our default choice is the SVD approach combined with AMP training. However, we observe that in certain models or extremely low-bit scenarios, numerical errors may occur within the AMP framework. In such cases, full-precision training becomes necessary.

| Training Recipe | | WikiText-2 PPL | C4 PPL | QA Acc | Memory | Time |
|---|---|---|---|---|---|---|
| FP32 | Inverse | 6.95 | 11.04 | 71.35 | 35384MiB | 2.2 hours |
| | SVD | 9.96 | 11.07 | 71.24 | 35360MiB | 2.2 hours |
| AMP | Inverse | 7.00 | 11.17 | 70.57 | 27624MiB | 0.9 hours |
| | SVD | 6.98 | 11.13 | 71.23 | 27554MiB | 0.9 hours |

Table 4: Comparison of different training recipes for FLATQUANT on the LLaMA-3-8B.

**Calibration Time.**   We further present the calibration time required by FLATQUANT for the LLaMA family in Table 5. Compared to SpinQuant (Liu et al., 2024c) and QAT methods, FLATQUANT requires significantly fewer computational resources and less training time, while delivering superior performance. For weight-only quantization, only transformations related to the linear weights are introduced, resulting in a shorter calibration time compared to weight-activation quantization. Moreover, as discussed in Section 4.2, FLATQUANT does not need to be combined with GPTQ to achieve optimal performance, further reducing the calibration overhead.

| LLaMA | 2-7B | 2-13B | 2-70B | 3-8B | 3-70B |
|---|---|---|---|---|---|
| weight-activation | 1.15 hours | 1.55 hours | 6.15 hours | 0.90 hours | 5.94 hours |
| weight-only | 0.67 hours | 1.01 hours | 5.00 hours | 0.70 hours | 4.89 hours |

Table 5: Calibration time for LLaMA models. The reported times correspond to training on 128 segments of 2048 tokens over 15 epochs with a batch size of 4, using a single GPU.

### B.2. Overhead Analysis of Affine Transformations

**Total FLOPs of Online Transformations.**   (1) Self-Attention. The self-attention module has three online transformations, i.e., $\{\mathbf{P}_a, \mathbf{P}_o, \mathbf{P}_h\}$. Suppose the hidden dimension $h_d$ and intermediate dimension $h_i$ of LLM can be perfectly decomposed into $\sqrt{h_d} \times \sqrt{h_d}$ and $\sqrt{h_i} \times \sqrt{h_i}$, respectively, then the total FLOPs of $\{\mathbf{P}_a, \mathbf{P}_o, \mathbf{P}_h\}$ is $4bsh_d\sqrt{h_d} + 2bsh_da + 4bsh_d^2/a$, where $b$ is the batch size, $s$ is the sequence length, and $a$ is the number of attention heads. (2) Feed-forward Network. The feed-forward module has two online transformations, i.e., $\{\mathbf{P}_{ug}, \mathbf{P}_d\}$. The total FLOPs of $\{\mathbf{P}_{ug}, \mathbf{P}_d\}$ is $4bsh_d\sqrt{h_d} + 4bsh_i\sqrt{h_i}$. In summary, the total FLOPs of the online transformations in a Transformer block amounts to $8bsh_d\sqrt{h_d} + 2bsh_da + 4bsh_d^2/a + 4bsh_i\sqrt{h_i}$. In LLaMA-2-7B (i.e., $h_d = 4096$, $h_i = 11008$ and $a = 32$), the FLOPs of online transformations only account for about $2.61\%$ of those of the FP16 model when $s$ reaches 2048.

---

[3] https://pytorch.org/docs/stable/generated/torch.nn.utils.parametrizations.orthogonal.html

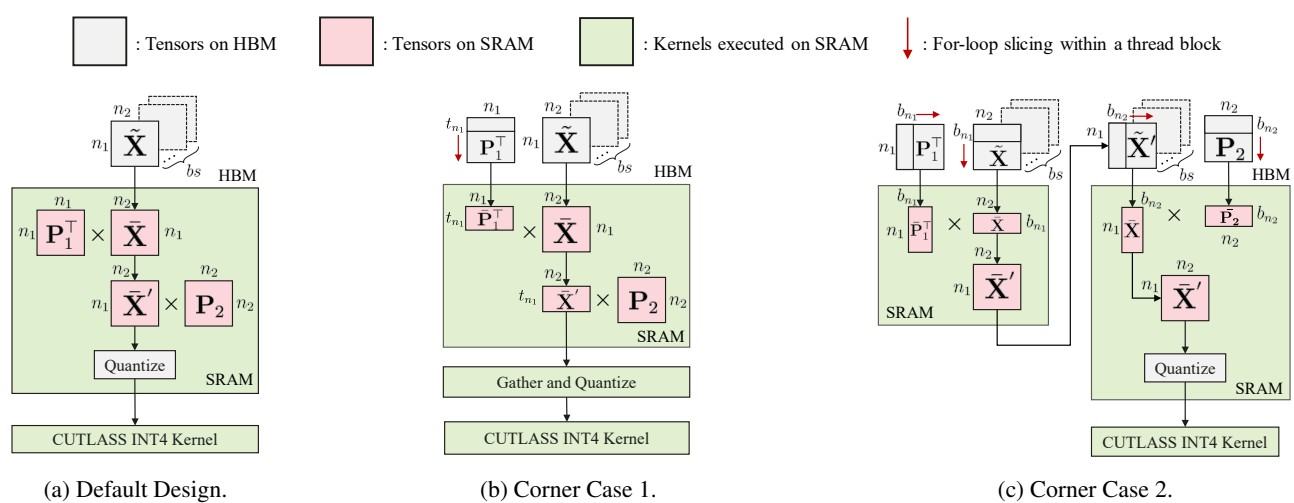

(a) Default Design.    (b) Corner Case 1.    (c) Corner Case 2.

Figure 8: The visualization of the kernel fusion in FLATQUANT based on the computation within a thread block. The design holds mainly for (a), where both transformations and quantization are fused together. For completeness, we also revise the design for corner cases in (b) and (c), when the SRAM is not large enough to hold the intermediate results.

**Memory Consumption of Online Transformations.** We compute the parameter count of each online transformation below: (1) $\mathbf{P}_a$: $2(\sqrt{h_d})^2$; (2) $\mathbf{P}_o$: $a^2$; (3) $\mathbf{P}_h$: $(h_d/a)^2$; (4) $\mathbf{P}_{ug}$: $2(\sqrt{h_d})^2$; (5) $\mathbf{P}_d$: $2(\sqrt{h_i})^2$. The total parameter count in one Transformer block is $4h_d + 2h_i + a^2 + (h_d/a)^2$. The additional memory consumption during inference is $2(4h_d + 2h_i + a^2 + (h_d/a)^2)$ bytes, which only consumes about 3.41MB extra memory space for LLaMA-2-7B.

### B.3. Detailed Design of Kernel Fusion

To avoid redundant memory access and improve computational efficiency, we attempt to fuse $\mathcal{Q}(\mathbf{P}_1^\top \times_1 \tilde{\mathbf{X}} \times_2 \mathbf{P}_2)$ into a single kernel, followed by the INT4 CUTLASS kernel to multiply the 4-bit quantized weights and activations. In most cases, the shared memory per thread block is sufficient to hold the source matrices $\mathbf{P}_1$, $\mathbf{P}_2$, $\bar{\mathbf{X}}$, and their intermediate results $\bar{\mathbf{X}}'$, as visualized in Figure 8a. Nonetheless, there are corner cases when the shared memory is insufficient to hold all necessary tensors (e.g., $n > 28762$ with $n_1, n_2 > 128$ on the NVIDIA RTX 3090). We thus revise our design for the two cases, as shown in Figure 8b and Figure 8c, respectively. To distinguish these scenarios more clearly, we have the following equations:

$$\textbf{Default Design:} \quad (n_1 * n_1 + 2 * n_1 * n_2) * 2 < m$$
$$(n_2 * n_2 + 2 * n_1 * n_2) * 2 < m \tag{5}$$
$$\textbf{Corner Case 1:} \quad (t_{n_1} * n_1 + n_1 * n_2 + t_{n_1} * n_2) * 2 < m$$
$$(n_2 * n_2 + 2 * t_{n_1} * n_2) * 2 < m \tag{6}$$
$$\textbf{Corner Case 2:} \quad (n_1 * b_{n_1} + b_{n_1} * n_2 + n_1 * n_2) * 2 < m$$
$$(n_1 * b_{n_2} + b_{n_2} * n_2 + n_1 * n_2) * 2 < m \tag{7}$$

where $m$ is the shared memory size per thread block, $t_{n_1}$ is the tiling size of non-reduction dimension of $\mathbf{P}_1$, $b_{n_1}$ is the tiling size of reduction dimension of $\mathbf{P}_1$, $b_{n_2}$ is the tiling size of reduction dimension of $\mathbf{P}_2$ and 2 refers to two bytes to hold tensors in float16. Below we review the designs for the two corner cases respectively.

**Corner Case 1.** When both $n$ and $n_1$ are excessively large, it is suggested to prevent from loading the entire $\mathbf{P}_1$ and $\bar{\mathbf{X}}$ into SRAM. We manage this by tiling the non-reduction dimension of $\mathbf{P}_1$ into $t_{n_1}$ slices. This strategy enables us to integrate $\bar{\mathbf{P}}_1\bar{\mathbf{X}}\mathbf{P}_2$ into one kernel, with $\bar{\mathbf{P}}_1$ representing a slice of $\mathbf{P}_1$ on the non-reduction dimension. Subsequently, we invoke a separate fused kernel for quantization, computing the quantization scale and scaling the input.

**Corner Case 2.** When both $n$ and $n_2$ are extremely large, $\mathbf{P}_1$, $\bar{\mathbf{X}}$ and $\mathbf{P}_2$ cannot be loaded into SRAM together. To handle this, we first compute $\bar{\mathbf{X}}' = \bar{\mathbf{P}}_1^\top \bar{\mathbf{X}}$, where each thread block slicing the non-reduction dimension of $\mathbf{P}_1$ and $\bar{\mathbf{X}}$ with the tiling shape $b_{n_1}$. The output $\tilde{\mathbf{X}}'$ is written back to the global memory, and the SRAM memory is thus released. Next, we

slice the non-reduction dimension of $\tilde{\mathbf{X}}^{'}$ and $\mathbf{P}_2$ with tiling size $b_{n_2}$, and compute the matrix multiplication, followed by quantizing the result on the fly.

**Kernel Profiling.**   We enumerate popular hidden sizes in the series of LLaMA models, and provide the detailed profiling results of FLATQUANT's online transformation with and without kernel fusion in Table 6. Note that the SRAM can hold all of these shapes with the default design on the NVIDIA RTX 3090. It can be found that kernel fusion achieves significant speedup across various hidden dimensions and batch sizes, e.g., 1.5x-3x prefill speedup and 1.2x-4x decoding speedup, respectively. We also selectively test the two corner cases with the hidden size of 28762, both of which bring considerably 2.3x speedup.

| Hidden Dimension | Batch Size | without Kernel Fusion | | with Kernel Fusion | | Speedup | |
|---|---|---|---|---|---|---|---|
| | | Prefill Time (ms) | Decode Time (ms) | Prefill Time (ms) | Decode Time (ms) | Prefill | Decode |
| | 1 | 0.1956 | 0.0184 | 0.0625 | 0.0082 | 3.13x | 2.25x |
| | 2 | 0.3809 | 0.0195 | 0.1116 | 0.0072 | 3.41x | 2.71x |
| | 4 | 0.7199 | 0.0212 | 0.2120 | 0.0082 | 3.40x | 2.59x |
| 4096 | 8 | 1.4019 | 0.0236 | 0.4188 | 0.0082 | 3.35x | 2.88x |
| | 16 | 2.7628 | 0.0307 | 0.8417 | 0.0073 | 3.28x | 4.20x |
| | 32 | 5.5101 | 0.0317 | 1.7091 | 0.0082 | 3.22x | 3.87x |
| | 64 | 10.9752 | 0.0328 | 3.4898 | 0.0082 | 3.14x | 4.00x |
| | 1 | 0.2519 | 0.0195 | 0.1321 | 0.0113 | 1.91x | 1.73x |
| | 2 | 0.4915 | 0.0205 | 0.2570 | 0.0113 | 1.91x | 1.82x |
| | 4 | 0.9073 | 0.0225 | 0.5161 | 0.0113 | 1.76x | 2.00x |
| 5120 | 8 | 1.7582 | 0.0266 | 1.0363 | 0.0113 | 1.70x | 2.36x |
| | 16 | 3.4748 | 0.0338 | 2.0480 | 0.0121 | 1.70x | 2.80x |
| | 32 | 6.9079 | 0.0358 | 4.1313 | 0.0123 | 1.67x | 2.92x |
| | 64 | 13.8619 | 0.0379 | 8.2033 | 0.0123 | 1.69x | 3.08x |
| | 1 | 0.3845 | 0.0195 | 0.1608 | 0.0132 | 2.39x | 1.48x |
| | 2 | 0.7393 | 0.0205 | 0.3092 | 0.0132 | 2.39x | 1.55x |
| | 4 | 1.4433 | 0.0205 | 0.6257 | 0.0123 | 2.31x | 1.67x |
| 8192 | 8 | 2.8529 | 0.0215 | 1.2411 | 0.0133 | 2.30x | 1.62x |
| | 16 | 5.6668 | 0.0225 | 2.4904 | 0.0133 | 2.28x | 1.69x |
| | 32 | 11.3183 | 0.0246 | 4.9418 | 0.0133 | 2.29x | 1.85x |
| | 64 | 22.6714 | 0.0297 | 9.8459 | 0.0143 | 2.30x | 2.07x |
| | 1 | 0.6154 | 0.0215 | 0.3830 | 0.0173 | 1.61x | 1.24x |
| | 2 | 1.2032 | 0.0225 | 0.7547 | 0.0173 | 1.59x | 1.30x |
| | 4 | 2.3654 | 0.0223 | 1.5032 | 0.0164 | 1.57x | 1.36x |
| 11008 | 8 | 4.7570 | 0.0236 | 2.9983 | 0.0174 | 1.59x | 1.35x |
| | 16 | 9.4536 | 0.0256 | 6.0099 | 0.0184 | 1.57x | 1.39x |
| | 32 | 18.9102 | 0.0287 | 12.0444 | 0.0195 | 1.57x | 1.47x |
| | 64 | 38.2700 | 0.0379 | 24.0000 | 0.0248 | 1.59x | 1.53x |
| | 1 | 0.7260 | 0.0225 | 0.4444 | 0.0184 | 1.63x | 1.22x |
| | 2 | 1.4203 | 0.0236 | 0.8653 | 0.0184 | 1.64x | 1.28x |
| | 4 | 2.8088 | 0.0246 | 1.7254 | 0.0184 | 1.63x | 1.33x |
| 13824 | 8 | 5.6228 | 0.0247 | 3.4273 | 0.0195 | 1.64x | 1.27x |
| | 16 | 11.2297 | 0.0266 | 6.8726 | 0.0195 | 1.63x | 1.37x |
| | 32 | 22.4302 | 0.0319 | 13.7216 | 0.0205 | 1.63x | 1.56x |
| | 64 | 45.4374 | 0.0471 | 27.4698 | 0.0275 | 1.65x | 1.72x |
| | 1 | 0.6932 | 0.0215 | 0.4178 | 0.0184 | 1.66x | 1.17x |
| | 2 | 1.3466 | 0.0225 | 0.8233 | 0.0184 | 1.64x | 1.22x |
| | 4 | 2.6557 | 0.0236 | 1.6507 | 0.0184 | 1.61x | 1.28x |
| 14336 | 8 | 5.2910 | 0.0246 | 3.2922 | 0.0195 | 1.61x | 1.26x |
| | 16 | 10.5185 | 0.0257 | 6.5966 | 0.0195 | 1.59x | 1.32x |
| | 32 | 20.9249 | 0.0317 | 13.0601 | 0.0205 | 1.60x | 1.55x |
| | 64 | 42.7981 | 0.0461 | 25.9308 | 0.0266 | 1.65x | 1.73x |

Table 6: Prefill and decoding speedup of kernel fusion across different hidden dimensions and batch sizes. The sequence length is 2048 for prefill and 1 for decoding. The default kernel design holds for all the above settings.

# C. Additional Experiments

## C.1. Results on Other LLM Architectures

**Results on LLaMA-3.1-8B-Instruct.** Aside from the pre-trained LLaMA models, we also investigate the quantization performance of LLaMA-3.1-8B-Instruct, a representative of the instruction-tuned LLM. The perplexity of language modeling and the accuracy of QA tasks for LLaMA-3.1-8B-Instruct are shown in Table 7, where FLATQUANT again outperforms QuaRot by a large margin.

| | WikiText-2 | C4 | ARC-C | ARC-E | HellaSwag | LAMBADA | PIQA | Winogrande | Avg |
|---|---|---|---|---|---|---|---|---|---|
| FP16 | 7.22 | 11.38 | 55.20 | 79.67 | 79.20 | 73.14 | 81.12 | 73.80 | 73.69 |
| QuaRot | 9.25 | 15.13 | 45.39 | 73.15 | 73.45 | 66.41 | 76.01 | 66.61 | 66.84 |
| FLATQUANT | 7.97 | 12.99 | 52.90 | 79.25 | 76.68 | 70.79 | 79.49 | 73.09 | **72.03** |

Table 7: Evaluation results of FLATQUANT on LLaMA-3.1-8B-Instruct.

**Results on Qwen-2.5-Instruct.** In addition to the series of LLaMA models, we also validate FLATQUANT on Qwen-2.5-Instruct, including both the 7B and 32B models. The results on language modeling and QA benchmarks are summarized in Table 8. For the 7B model, FLATQUANT achieved a slightly lower average performance compared to the FP16 baseline, with an average score of 68.62. For the 32B model, FLATQUANTalso demonstrates competitive performance, achieving an average score of 74.89 (e.g., merely 0.21% drop), which is slightly lower than the FP16 baseline but higher than the QuaRot.

| Model | Method | W Quantizer | WikiText-2 | C4 | ARC-C | ARC-E | HellaSwag | LAMBADA | PIQA | Winogrande | Avg |
|---|---|---|---|---|---|---|---|---|---|---|---|
| **7B** | FP16 | - | 8.36 | 14.37 | 51.37 | 75.80 | 79.57 | 67.61 | 80.20 | 69.93 | 70.75 |
| | FLATQUANT | RTN | 8.46 | 13.94 | 51.71 | 77.69 | 78.42 | 57.46 | 76.93 | 69.53 | **68.62** |
| **32B** | FP16 | - | 5.32 | 10.45 | 58.62 | 77.02 | 85.25 | 75.14 | 81.39 | 73.16 | 75.10 |
| | QuaRot | RTN | 6.95 | 12.17 | 52.13 | 74.37 | 80.41 | 68.37 | 78.45 | 67.72 | 70.24 |
| | QuaRot | GPTQ | 6.54 | 11.65 | 56.06 | 76.52 | 81.83 | 71.26 | 78.78 | 69.06 | 72.25 |
| | FLATQUANT | RTN | 5.80 | 10.86 | 58.62 | 78.58 | 83.72 | 75.26 | 80.74 | 72.45 | **74.89** |

Table 8: Evaluation results of FLATQUANT on Qwen-2.5-Instruct models.

**Results on DeepSeek-V3-Base and DeepSeek-R1.** We further scale the evaluation of FLATQUANT to DeepSeek-V3-Base and DeepSeek-R1, both of which are large-scale Mixture-of-Experts (MoE) models of 671B parameters. Table 9 presents the results under 4-bit weight and activation quantization. It can be found that FLATQUANT maintains strong performance across both LLMs, demonstrating its applicability beyond standard dense LLMs.

| Model | Quantization | C-Eval | MMLU | AIME2024 |
|---|---|---|---|---|
| **DeepSeek V3-Base** | FP8 | 90.10 | 87.10 | - |
| | FLATQUANT-W4A4 | 89.59 | 86.32 | - |
| **DeepSeek R1** | FP8 | - | - | 79.8 |
| | FLATQUANT-W4A4 | - | - | 73.3 |

Table 9: Evaluation results of FLATQUANT on LLaMA-3.1-8B-Instruct.

## C.2. Results on MT-Bench

Aside from language modeling and question answering, we also evaluate FLATQUANT on MT-Bench with LLaMA-3.1-8B-Instruct model in Table 10. We use GPT-4o as the evaluator to justify the ability multi-turn conversation. It can be found that while FLATQUANT trails behind the FP16 baseline in coding and STEM, it consistently outperforms QuaRot with GPTQ across all categories, narrowing the gap between the quantized model and the FP16 baseline. Notably, for math problems, FLATQUANT matches the FP16 baseline's score, exceeding QuaRot by 1.9 points.

| Method | Writing | Roleplay | Reasoning | Math | Coding | Extraction | STEM | Humanities | Avg |
|---|---|---|---|---|---|---|---|---|---|
| FP16 | 8.17 | 8.10 | 5.05 | 7.00 | 6.10 | 8.67 | 8.50 | 8.91 | 7.60 |
| QuaRot | 7.20 | 6.90 | 3.90 | 5.30 | 4.05 | 6.70 | 6.05 | 7.80 | 5.99 |
| FLATQUANT | 7.95 | 7.35 | 4.70 | 7.20 | 4.80 | 7.60 | 7.20 | 8.70 | **6.94** |

Table 10: MT-Bench results of 4-bit weight & activation quantized LLaMA-3.1-8B-Instruct model.

## C.3. Extension to More Quantization Settings

**Weight-only Quantization**    While our primary analysis focuses on joint full weight-activation-kvcache quantization schemes (Section 4.2), FLATQUANT demonstrates notable versatility across different quantization paradigms. Table 11 presents the results of the weight-only quantization compared to several state-of-the-art baselines in 4 bits and 3 bits. We adopt per-channel symmetric quantization for weights, maintaining consistency with our full quantization scheme's weight processing methodology. FLATQUANT again obtains leading accuracy compared with leading baselines. Specifically, it outperforms RTN, GTPQ, and AWQ, while also slightly surpassing GPTQ with per-group quantization (group size = 128). Meanwhile, FLATQUANT performs comparably to QuIP. Additionally, we report results for FLATQUANT when combined with GPTQ, where GPTQ utilizes the same calibration data for both closed-form weight updates and training. The findings remain consistent with the results in joint full weight-activation-kvcache quantization schemes.

| LLaMA-3-8B | WikiText-2 PPL | | C4 PPL | |
|---|---|---|---|---|
| | W4A16 | W3A16 | W4A16 | W3A16 |
| FP16 | 6.14 | | 9.45 | |
| RTN | 8.70 | 2.2E3 | 14.00 | 5.6E3 |
| GPTQ | 7.00 | 13.00 | 11.80 | 45.90 |
| GPTQ-g128 | 6.50 | 8.20 | 10.40 | 13.70 |
| AWQ | 7.10 | 12.80 | 10.10 | 16.80 |
| QuIP | 6.50 | 7.50 | 11.10 | 11.30 |
| FLATQUANT-RTN | 6.54 | 7.78 | 10.17 | 12.64 |
| FLATQUANT-GPTQ | 6.48 | 7.52 | 10.28 | 12.91 |

Table 11: WikiText-2 and C4 perplexity of weight-only quantizationon on LLaMA-3-8B model.

| K bits | V bits | WikiText-2 | C4 | ARC-C | ARC-E | HellaSwag | LAMBADA | PIQA | Winogrande | Avg |
|---|---|---|---|---|---|---|---|---|---|---|
| 16 | 16 | 6.14 | 9.45 | 53.50 | 77.57 | 79.12 | 75.51 | 80.74 | 72.93 | 73.23 |
| 4 | 4 | 6.20 | 9.56 | 52.82 | 78.20 | 79.13 | 75.32 | 80.47 | 72.77 | 73.12 |
| 4 | 3 | 6.25 | 9.66 | 52.90 | 77.65 | 79.00 | 75.10 | 80.79 | 73.48 | 73.15 |
| 4 | 2 | 6.60 | 10.33 | 49.32 | 74.37 | 77.88 | 72.77 | 79.22 | 72.69 | 71.04 |
| 3 | 4 | 6.35 | 9.91 | 52.05 | 77.95 | 78.41 | 73.94 | 79.71 | 73.48 | 72.59 |
| 3 | 3 | 6.41 | 10.03 | 52.47 | 76.85 | 78.25 | 74.02 | 79.98 | 72.61 | 72.36 |
| 3 | 2 | 6.84 | 10.83 | 47.44 | 73.91 | 77.18 | 70.37 | 78.73 | 71.19 | 69.80 |
| 2 | 4 | 7.70 | 13.36 | 49.15 | 74.62 | 74.74 | 63.65 | 77.58 | 68.67 | 68.07 |
| 2 | 3 | 7.79 | 13.44 | 46.67 | 71.63 | 74.17 | 63.05 | 77.48 | 68.51 | 66.92 |
| 2 | 2 | 8.93 | 16.13 | 42.92 | 68.60 | 71.54 | 55.58 | 75.30 | 64.40 | 63.06 |

Table 12: Different bits for KV cache quantization on the LLaMA-3-8B model.

**KV Cache Quantization.**    To further evaluate its versatility, we apply FLATQUANT to KV cache only quantization. In this setting, we retain high precision for the rest of the model (including weights and activations) and apply the group-wise asymmetric quantization (with a group size of 128) to keys and values. Table 12 presents the results of KV cache quantization using various bit-widths on the LLaMA-3-8B model. Consistent with previous studies (Hooper et al., 2024; Liu et al., 2024b; Ashkboos et al., 2024), we observe that keys are more sensitive to quantization than values. Furthermore, Table 13

| Methods | K bits | V bits | LLaMA-2-7B | LLaMA-2-13B |
|---|---|---|---|---|
| | 16 | 16 | 5.47 | 4.88 |
| QuaRot | 4 | 4 | 5.51 | 4.91 |
| | 3 | 3 | 5.68 | 5.02 |
| | 2 | 2 | 9.23 | 7.07 |
| FLATQUANT | 4 | 4 | 5.50 | 4.91 |
| | 3 | 3 | 5.61 | 5.00 |
| | 2 | 2 | 6.66 | 5.69 |

Table 13: WikiText-2 perplexity of LLaMA-2 models with different bits of KV cache quantization.

compares FLATQUANT with QuaRot for KV cache quantization on LLaMA-2-7B and LLaMA-2-13B models. As shown, FLATQUANT delivers superior performance in most cases, particularly for lower-bit (2-3 bits). When both keys and values are quantized to 2 bits, FLATQUANT outperforms QuaRot by 2.57 in perplexity for the 7B model.

**Extreme Low-bit Quantization.** We quantize the LLM to extreme low-bit representations (e.g., INT3) to investigate the limitations of quantization. The results in Table 14 show that FLATQUANT still keeps most of the model's abilities in the 3-bit setting, whereas QuaRot struggles under such extreme low-bit conditions. Nevertheless, 4-bit quantization remains a better balance between inference resource efficiency and acceptable performance degradation for now.

| LLaMA3-8B | WikiText-2 | C4 | ARC-C | ARC-E | HellaSwag | LAMBADA | PIQA | Winogrande | Avg |
|---|---|---|---|---|---|---|---|---|---|
| FP16 | 6.14 | 9.45 | 53.50 | 77.57 | 79.12 | 75.51 | 80.74 | 72.93 | 73.23 |
| QuaRot-W4A4KV4 | 8.16 | 13.38 | 45.73 | 70.83 | 72.97 | 62.70 | 75.35 | 67.17 | 65.79 |
| FLATQUANT-W4A4KV4 | 6.98 | 11.13 | 50.00 | 75.80 | 76.80 | 72.91 | 79.16 | 72.69 | **71.23** |
| QuaRot-W3A3KV3 | 686.54 | 630.89 | 25.34 | 28.41 | 28.07 | 0.78 | 50.71 | 48.70 | 30.33 |
| FLATQUANT-W3A3KV3 | 10.82 | 19.03 | 35.41 | 63.26 | 65.30 | 52.49 | 73.56 | 60.69 | **58.45** |

Table 14: Extreme low bit quantization results on LLAMA-3-8B models.

**Train One and Get More.** Remarkably, we demonstrate that the affine transformations learned from weight-activation quantization can be directly applied to other quantization settings, such as weight-only or KV cache quantization, with surprisingly strong performance. The associated results are presented in Table 15. For instance, the results labeled as "W4" are comparable to those in Table 11 that are specifically trained for weight-only quantization. This significantly saves time when applying FLATQUANT to different quantization settings, as only one set of transformation matrices is saved.

| W4 | A4 | KV4 | WikiText-2 PPL | C4 PPL | QA Acc |
|---|---|---|---|---|---|
| | | | 6.14 | 9.45 | 73.23 |
| ✓ | | | 6.56 | 10.25 | 72.92 |
| | ✓ | | 6.49 | 10.13 | 72.20 |
| | | ✓ | 6.23 | 9.61 | 73.43 |
| ✓ | ✓ | ✓ | 6.98 | 11.13 | 71.23 |

Table 15: Extending the affine transformations trained under W4A4KV4 to different quantization settings on LLaMA-3-8B model. QA Acc is the average accuray of the six QA tasks in lm-eval-harness.

## C.4. Detailed Ablation Study

To better disentangle the contributions of affine transformations and clipping thresholds, we present the full ablation results in Table 16, as an extension of the analysis in Table 3. The results highlight the effectiveness of each component in FLATQUANT, with learnable transformations (LT) playing a central role. Moreover, when built upon LT, other components further enhance the overall performance of FLATQUANT.

| LT | PS | LCT | WikiText-2 | C4 | ARC-C | ARC-E | HellaSwag | LAMBADA | PIQA | Winogrande | Avg. |
|----|----|-----|-----------|------|-------|-------|-----------|---------|------|------------|------|
| | | | 1266.60 | 936.41 | 25.26 | 28.62 | 27.04 | 1.26 | 51.80 | 51.93 | 30.99 |
| | ✓ | | NaN | NaN | 22.70 | 25.08 | 25.04 | 0.00 | 49.51 | 49.57 | 28.65 |
| | | ✓ | 1149.08 | 1490.08 | 22.95 | 29.29 | 27.35 | 0.60 | 52.99 | 50.83 | 30.67 |
| | ✓ | ✓ | 8197.96 | 4654.07 | 25.43 | 25.72 | 25.96 | 0.02 | 50.49 | 48.86 | 29.41 |
| ✓ | | | 8.50 | 13.51 | 44.97 | 71.38 | 73.17 | 67.05 | 76.88 | 67.48 | 66.82 |
| ✓ | ✓ | | 7.95 | 12.74 | 44.20 | 71.89 | 74.21 | 68.72 | 77.15 | 66.30 | 67.08 |
| ✓ | | ✓ | 7.11 | 11.47 | 49.32 | 76.14 | 76.30 | 72.17 | 78.89 | 71.51 | 70.72 |
| ✓ | ✓ | ✓ | 6.98 | 11.13 | 50.00 | 75.80 | 76.80 | 72.91 | 79.16 | 72.69 | 71.23 |

Table 16: Ablation study of FLATQUANT's main components on LLaMA-3-8B.

## C.5. Additional Discussions

**Calibration Set.** Since FLATQUANT employs a gradient-based method to optimize transformations for increased flatness, one reasonable concern is whether FLATQUANT might overfit the calibration set. To assess its generalization ability, we conducted an ablation study using different calibration datasets: WikiText-2, C4, and Pile. As shown in Table 17, FLATQUANT maintains stable performance across all datasets. For example, when calibrated on different datasets, FLATQUANT exhibits similar performance on WikiText-2, with PPL ranging from 6.98 to 7.04. On the C4 dataset, results are equally consistent, with PPLs between 11.05 and 11.13. Furthermore, QA accuracy remains within a narrow range (71.04% to 71.23%), suggesting that FLATQUANT generalizes well across different calibration datasets. This robustness is attributed to FLATQUANT's focus on learning an equivalent affine transformation with minimal quantization loss, rather than altering the model's weights. Nevertheless, it is reasonable to assume that the diversity of calibration data can further enhance the performance of our method.

| Calibration set | WikiText-2 | C4 | ARC-C | ARC-E | HellaSwag | LAMBADA | PIQA | Winogrande | Avg |
|-----------------|-----------|------|-------|-------|-----------|---------|------|------------|-----|
| WikiText2 | 6.98 | 11.13 | 50.00 | 75.80 | 76.80 | 72.91 | 79.16 | 72.69 | 71.23 |
| C4 | 7.04 | 11.05 | 50.34 | 75.38 | 76.74 | 73.28 | 78.67 | 71.82 | 71.04 |
| Pile | 7.04 | 11.08 | 51.11 | 77.36 | 76.63 | 72.37 | 78.94 | 70.56 | 71.16 |

Table 17: Ablation study of FLATQUANT's calibration set on LLaMA-3-8B model.

**Effect of Clipping.** While weight clipping has been widely adopted in LLM quantization, activation clipping remains relatively underexplored. Prior works (Ashkboos et al., 2024; Liu et al., 2024c) have shown that activation clipping alone yields limited benefits, primarily due to the presence of severe outliers. In contrast, our method demonstrates that learnable clipping thresholds (LCT), when applied after transformation, yields significant improvements. As shown in Table 18, applying LCT before transformation, similar to the RTN-style approach, yields only marginal gains. This observation is consistent with prior findings (Dettmers et al., 2022), suggesting that early clipping fails to effectively suppress activation outliers. In our method, the affine transformation redistributes outliers across channels, enabling the subsequent LCT step to more effectively clip a larger portion of extreme values. Crucially, the inverse transformation retains the ability to recover the original scale of informative signals, thereby preserving model quality after quantization. For comparison, we also report results using a QuaRot-style clipping strategy (with thresholds of 0.9 for activations and 0.95 for KV cache values). Overall, these results highlight that the integration of learnable transformations and adaptive clipping significantly enhances the performance of weight-activation quantization.

| LLaMA3-8B | WikiText-2 | C4 | ARC-C | ARC-E | HellaSwag | LAMBADA | PIQA | Winogrande | Avg |
|-----------|-----------|------|-------|-------|-----------|---------|------|------------|-----|
| FP16 | 6.14 | 9.45 | 53.50 | 77.57 | 79.12 | 75.51 | 80.74 | 72.93 | 73.23 |
| w/o LCT | 7.95 | 12.74 | 44.20 | 71.89 | 74.21 | 68.72 | 77.15 | 66.30 | 67.08 |
| LCT before Transformation | 7.37 | 11.86 | 48.72 | 76.18 | 75.11 | 66.65 | 77.91 | 67.17 | 68.62 |
| QuaRot-style Fixed Threshold | 7.25 | 11.62 | 48.21 | 75.29 | 75.66 | 71.32 | 78.73 | 70.01 | 69.87 |
| LCT after Transformation | 6.98 | 11.13 | 50.00 | 75.80 | 76.80 | 72.91 | 79.16 | 72.69 | 71.23 |

Table 18: The effect of Learnable Clipping Thresholds.

**Effect of Mixed-precision Schemes.** To further evaluate the practicality of FlatQuant, we evaluate its performance under a mixed-precision quantization scheme. Specifically, we explore how integrating FlatQuant into a layer-wise heterogeneous bit-width setting can improve accuracy while retaining high inference speedup. As shown in Table 19, we apply W8A8 quantization to the top 5 important Transformer layers and all down-projection layers, based on their relative importance (Kim et al., 2024). This configuration substantially mitigates the degradation observed in uniform low-bit settings. These results indicate that FlatQuant can be effectively combined with mixed-precision strategies, enhancing its applicability in real-world deployments.

| | WikiText-2 | C4 | ARC-C | ARC-E | HellaSwag | LAMBADA | PIQA | Winogrande | Avg |
|---|---|---|---|---|---|---|---|---|---|
| FP16 | 6.14 | 9.45 | 53.50 | 77.57 | 79.12 | 75.51 | 80.74 | 72.93 | 73.23 |
| FLATQUANT | 6.98 | 11.13 | 50.00 | 75.80 | 76.80 | 72.91 | 79.16 | 72.69 | 71.23 |
| + down_proj_8bits | 6.73 | 10.62 | 50.00 | 77.78 | 77.49 | 73.96 | 79.54 | 70.56 | 71.55 |
| + Top5_8bits | 6.80 | 10.82 | 49.23 | 76.56 | 76.54 | 73.51 | 79.71 | 73.95 | 71.58 |
| + Top5 & down_proj_8bits | 6.61 | 10.43 | 51.11 | 77.74 | 77.47 | 74.69 | 79.92 | 72.14 | 72.18 |

Table 19: The effect of mixed-precision quantization with selectively higher-bit layers in FlatQuant.

## C.6. Experiment Details of Figure 5

In Figure 5, we present the prefill speedup and WikiText2 PPL results of different decomposed matrix sizes on LLaMA-2-7B model. We decompose the hidden dimension 4096 into $n_1 \times n_2$ and range $n_1$ from 1 to 2048, where $n_1 = 1$ amounts to maintaining a full-size transformation matrix. The intermediate dimension 11008 is decomposed into $64 \times 172$ as done in FLATQUANT. For PPL evaluation, we only quantize the last Transformer block and learn the affine transformations within it. For speedup evaluation, we do not leverage the online transformation kernel in Section 3.3 and implement online transformations with naive matrix multiplication in PyTorch.

## C.7. Analysis of Inference Memory

To further validate the efficiency of FLATQUANT, we provide additional results on its inference-time memory consumption. These experiments complement our theoretical analysis in Appendix B.2, and empirically confirm that the online affine transformations introduced in FlatQuant incur negligible memory overhead.

Table 20 reports the peak memory usage during single-token decoding on a single Transformer layer of the LLaMA-2-7B model, under varying KV cache lengths, with batch size set to 1. We compare standard FP16 inference, INT4 quantization, and our method, FlatQuant. As shown, FlatQuant matches the memory efficiency of INT4 quantization across all sequence lengths, achieving a consistent memory reduction of over 3.3× compared to FP16. These results indicate that FlatQuant preserves the low memory footprint of INT4 quantization, while introducing no additional memory cost from its online processing.

| Sequence Length | FP16 (GB) | INT4 (GB) | FlatQuant (GB) | Saving Factor |
|---|---|---|---|---|
| 256 | 0.393 | 0.110 | 0.110 | 3.58× |
| 512 | 0.399 | 0.112 | 0.112 | 3.56× |
| 1024 | 0.411 | 0.118 | 0.118 | 3.48× |
| 2048 | 0.434 | 0.130 | 0.130 | 3.35× |

Table 20: Peak memory usage for decoding a single token on one Transformation layer of LLaMA-2-7B model with KV caches of different lengths and batch size 1.

## C.8. Additional Analysis of Inference Latency

**Baseline.** We implement and report the latency results of INT4 quantization and QuaRot with QuaRot's official code[4]. These baselines share the same quantization settings with FLATQUANT as described in Section 4.1 for fair comparison.

---

[4]https://github.com/spcl/QuaRot

**End-to-end Speedup.** We decode 256 tokens after the prefill on a sequence length of 2048 and provide the prefill and decoding speedup of FLATQUANT in Figure 9 and Figure 10. FLATQUANT achieves a prefill speedup of 2.30x and decoding speedup of 1.76x under the batch size of 64, with only 0.07x speedup loss compared to the naive INT4 quantization for both prefill and decoding. Note that when the batch size is smaller than 16, quantization overhead outweighs the benefits brought by KV cache memory reduction for the decoding stage, resulting in less than 1x speedup for both INT4 quantization and FLATQUANT. However, since the decoding speedup shows good scalability with the batch size, we can gain a practical decoding speedup simply by employing a large batch size.

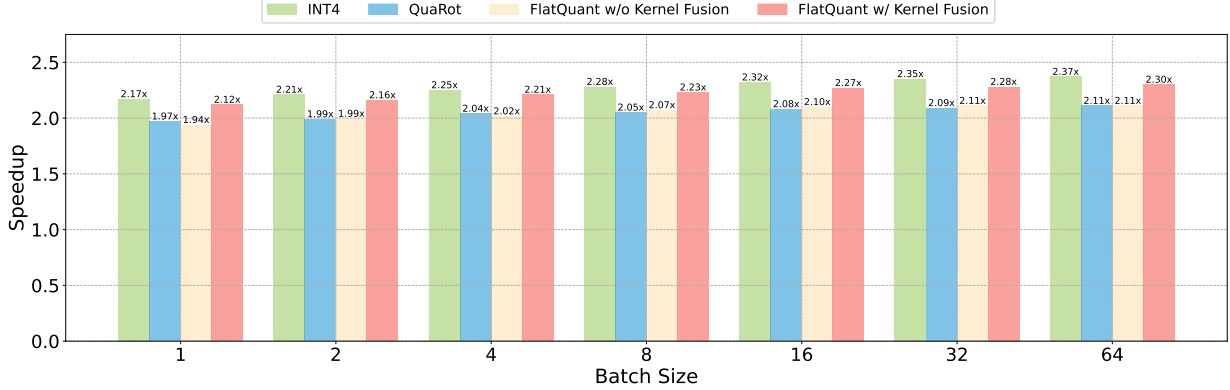

Figure 9: Prefill speedup of LLaMA-2-7B on a sequence length of 2048.

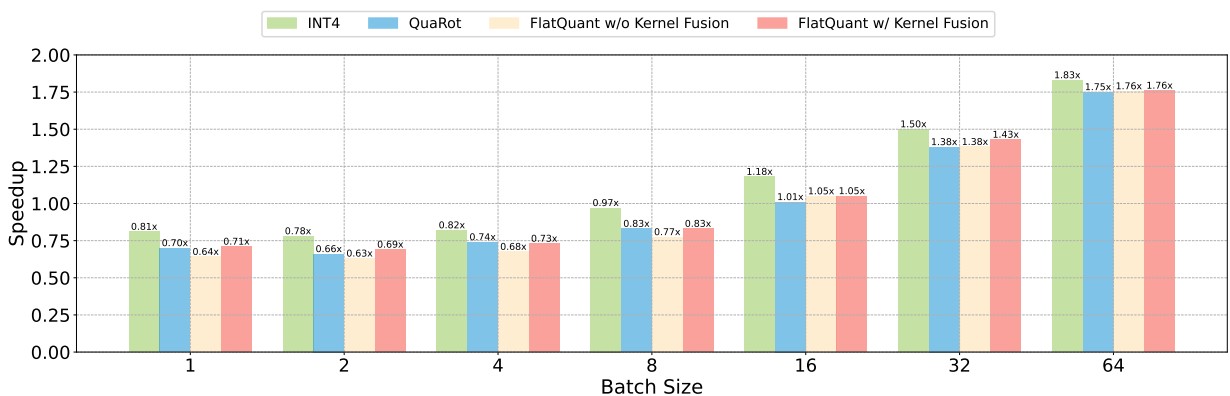

Figure 10: Decoding speedup on LLaMA-2-7B model. We decode 256 tokens after the prefill on a sequence length of 2048.

**Speedup across Sequence Lengths.** In addition to the results presented in Figure 4, we further evaluate the runtime speedup of FlatQuant under varying sequence lengths to better characterize its practical efficiency. Table 21 and Table 22 provide detailed prefill and decoding speedups on LLaMA-3-8B, under different input and KV cache lengths. For the prefill stage with batch size 1, FLATQUANT achieves consistent speedups across sequence lengths, achieving 2.12× at length 2048 and 1.80× at length 16384, comparable to INT4 and outperforming QuaRot. In the decoding stage with batch size 64, FLATQUANT consistently surpasses QuaRot across all KV cache lengths and closely approaches the efficiency of INT4 quantization. These results demonstrate that FLATQUANT maintains its low-overhead advantage across a wide range of generation scenarios, including both short and long contexts.

### C.9. Comparison with AffineQuant

AffineQuant also proposed learning an equivalent affine transformation to reduce quantization error. As shown in Table 1, AffineQuant and FLATQUANT exhibit markedly different performance. To further clarify why FLATQUANT outperforms AffineQuant, how their differences affect quantization performance, and what factors matter most in quantization, we

| Prefill Length | INT4 | QuaRot | FlatQuant |
|---|---|---|---|
| 2048 | 2.16× | 1.97× | 2.12× |
| 4096 | 2.06× | 1.90× | 2.04× |
| 8192 | 1.94× | 1.79× | 1.92× |
| 16384 | 1.83× | 1.72× | 1.80× |

Table 21: Prefill Speedup on LLaMA-3-8B Compared to FP16 for Different Input Sequence Lengths at Batch Size 1.

| KV Cache Length | INT4 | QuaRot | FlatQuant |
|---|---|---|---|
| 256 | 1.38× | 1.09× | 1.24× |
| 512 | 1.62× | 1.38× | 1.56× |
| 1024 | 1.70× | 1.61× | 1.63× |
| 2048 | 1.78× | 1.72× | 1.76× |

Table 22: Decoding Speedup on LLaMA-3-8B Compared to FP16 for Different KV Cache Lengths at Batch Size 64.

provide additional analysis below.

**Improved Expressivity.** Although FLATQUANT adopts a Kronecker product with two lightweight matrices, its expressivity remains competitive. As shown in Figure 5, the decomposed transformation achieves accuracy comparable to a full-size affine transformation, suggesting that such a structural constraint does not lead to a loss in functional capacity. In contrast, AffineQuant enforces strictly diagonally dominant transformations to ensure invertibility, which may inadvertently reduce expressivity and make the transformation behave similarly to per-channel scaling (see Figure 7 in the AffineQuant paper).

**Applicability to All Linear Layers.** FLATQUANT offers broader applicability across the model architecture. Thanks to its lightweight Kronecker structure, it can be efficiently applied to all linear layers with negligible overhead. In contrast, AffineQuant directly learns a full-size affine matrices, which are only feasible to apply to output projection layers due to their high computational cost. For other layers, AffineQuant reverts to simpler per-channel scaling. These practical limitations further constrain its overall effectiveness, whereas FLATQUANT maintains a unified and expressive transformation across the entire model.

# D. Additional Visualizations

## D.1. More Visualizations of Weight and Activation Distributions

**Experiment Details.** We visualize the distribution of weights and activations after different transformations, including per-channel scaling in SmoothQuant (Xiao et al., 2023), Hadamard transformation in QuaRot (Ashkboos et al., 2024), and affine transformation in FLATQUANT. We compute the per-channel Frobenius norm to quantify the channel magnitude. We randomly sample from the C4 (Raffel et al., 2020) dataset to collect activation statistics.

**Visualizations on the LLaMA Models.** We visualize the distribution envelopes of both original and transformed weights and activations on the LLaMA models in Figure 11-15. It can be observed that neither per-channel scaling nor Hadamard transformation can fully smooth out outlier channels to produce flatness, still leaving outlier channels, especially on activations. On the other hand, the affine transformation learned by FLATQUANT can effectively produce flatter distributions for both weights and activations which are easier to quantize.

## D.2. More Visualizations of Quantization Error Landscapes

**Experiment Details.** We randomly sample 128 samples from the C4 (Raffel et al., 2020) dataset and compute their average mean squared error for visualization. For per-channel scaling, we follow SmoothQuant (Xiao et al., 2023) and only perform per-channel scaling for the inputs of the self-attention and feed-forward modules. For the Hadamard transformation, we replace the affine transformation in FLATQUANT with a fixed Hadamard transformation. The quantization settings are the same as those described in Section 4.1.

**Visualizations on the LLaMA Models.** We visualize the quantization error landscapes of LLaMA models in Figure 2 and Figure 18-21. With the affine transformation to smooth outliers, FLATQUANT can effectively suppress the quantization errors at pivot tokens and ease the quantization error propagation, leading to a flatter quantization error landscape compared with per-channel scaling and Hadamard transformation.

# E. Limitations

In this study, we present FLATQUANT, but there are certain limitations to acknowledge. First, the full potential of 4-bit quantization has not been thoroughly explored. While we follow the previous studies to build the calibration set and demonstrate that FLATQUANT is robust across various data sources, the optimal selection of calibration sets remains an open question. Additionally, our focus has primarily been on the INT4 data type, and we have not examined the integration of FLATQUANT with newer data types, such as MXFP4, which may offer advantages over INT4. Addressing these aspects represents promising avenues for future research.

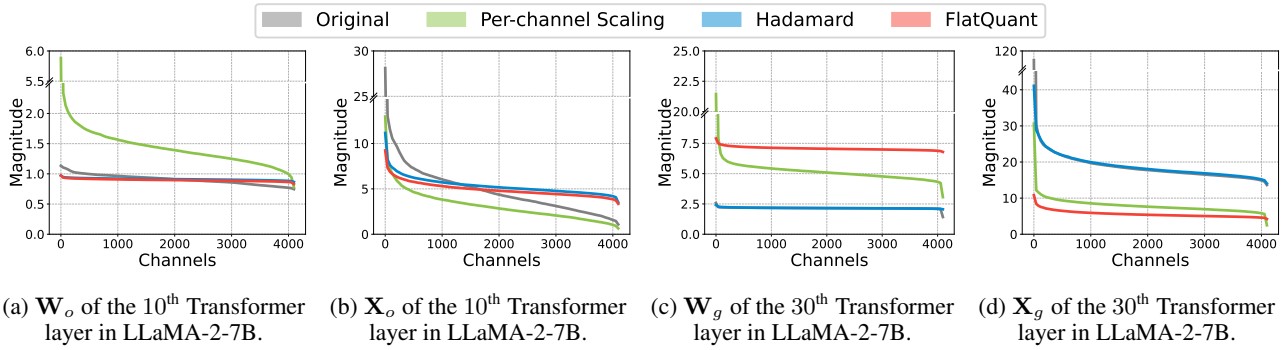

(a) $\mathbf{W}_o$ of the 10th Transformer layer in LLaMA-2-7B.

(b) $\mathbf{X}_o$ of the 10th Transformer layer in LLaMA-2-7B.

(c) $\mathbf{W}_g$ of the 30th Transformer layer in LLaMA-2-7B.

(d) $\mathbf{X}_g$ of the 30th Transformer layer in LLaMA-2-7B.

Figure 11: Distributions of weights and inputs from LLaMA-2-7B, sorted by the channel magnitudes (i.e., the Frobenius norm) in descending order.

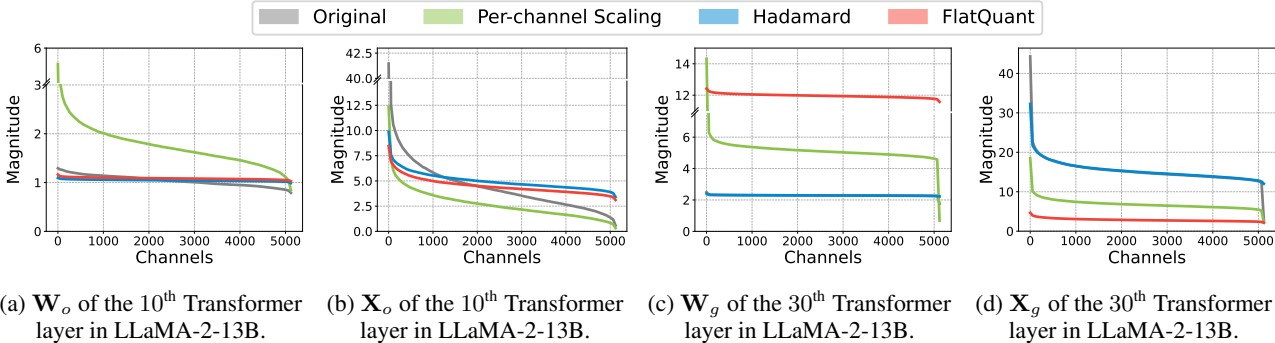

(a) $\mathbf{W}_o$ of the 10th Transformer layer in LLaMA-2-13B.

(b) $\mathbf{X}_o$ of the 10th Transformer layer in LLaMA-2-13B.

(c) $\mathbf{W}_g$ of the 30th Transformer layer in LLaMA-2-13B.

(d) $\mathbf{X}_g$ of the 30th Transformer layer in LLaMA-2-13B.

Figure 12: Distributions of weights and inputs from LLaMA-2-13B, sorted by the channel magnitudes (i.e., the Frobenius norm) in descending order.

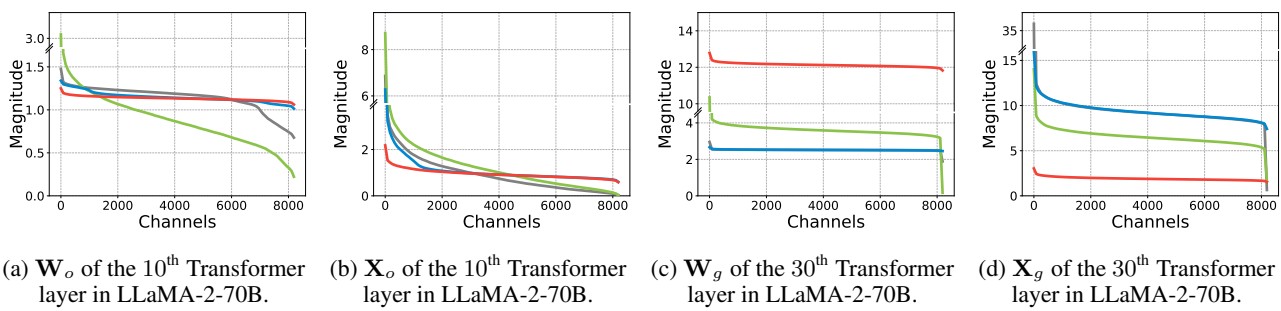

(a) $\mathbf{W}_o$ of the 10th Transformer layer in LLaMA-2-70B.

(b) $\mathbf{X}_o$ of the 10th Transformer layer in LLaMA-2-70B.

(c) $\mathbf{W}_g$ of the 30th Transformer layer in LLaMA-2-70B.

(d) $\mathbf{X}_g$ of the 30th Transformer layer in LLaMA-2-70B.

Figure 13: Distributions of weights and inputs from LLaMA-2-70B, sorted by the channel magnitudes (i.e., the Frobenius norm) in descending order.

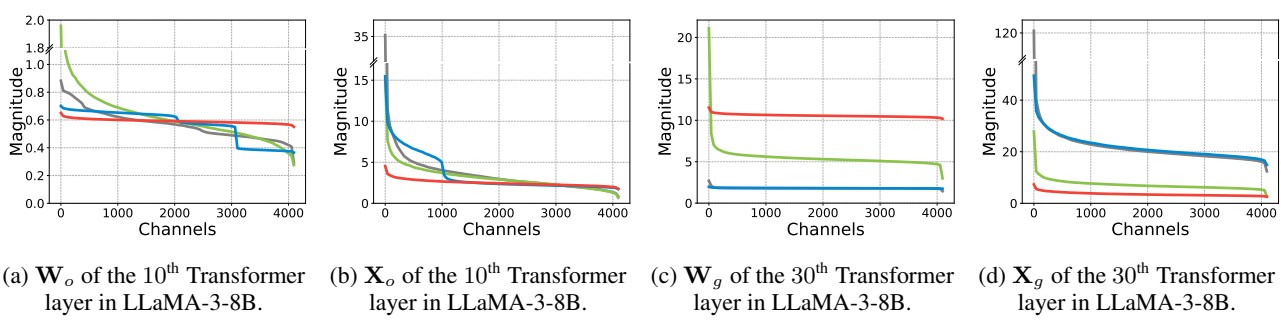

(a) $\mathbf{W}_o$ of the 10th Transformer layer in LLaMA-3-8B.

(b) $\mathbf{X}_o$ of the 10th Transformer layer in LLaMA-3-8B.

(c) $\mathbf{W}_g$ of the 30th Transformer layer in LLaMA-3-8B.

(d) $\mathbf{X}_g$ of the 30th Transformer layer in LLaMA-3-8B.

Figure 14: Distributions of weights and inputs from LLaMA-3-8B, sorted by the channel magnitudes (i.e., the Frobenius norm) in descending order.

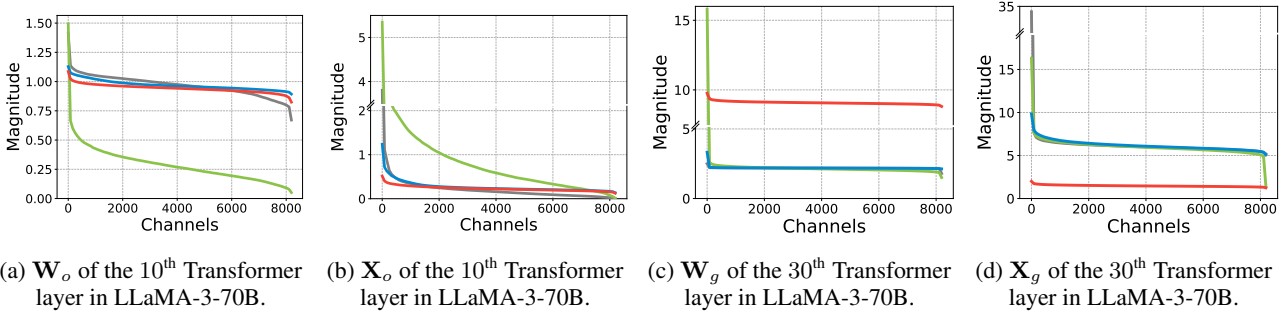

(a) $\mathbf{W}_o$ of the 10th Transformer layer in LLaMA-3-70B.

(b) $\mathbf{X}_o$ of the 10th Transformer layer in LLaMA-3-70B.

(c) $\mathbf{W}_g$ of the 30th Transformer layer in LLaMA-3-70B.

(d) $\mathbf{X}_g$ of the 30th Transformer layer in LLaMA-3-70B.

Figure 15: Distributions of weights and inputs from LLaMA-3-70B, sorted by the channel magnitudes (i.e., the Frobenius norm) in descending order.

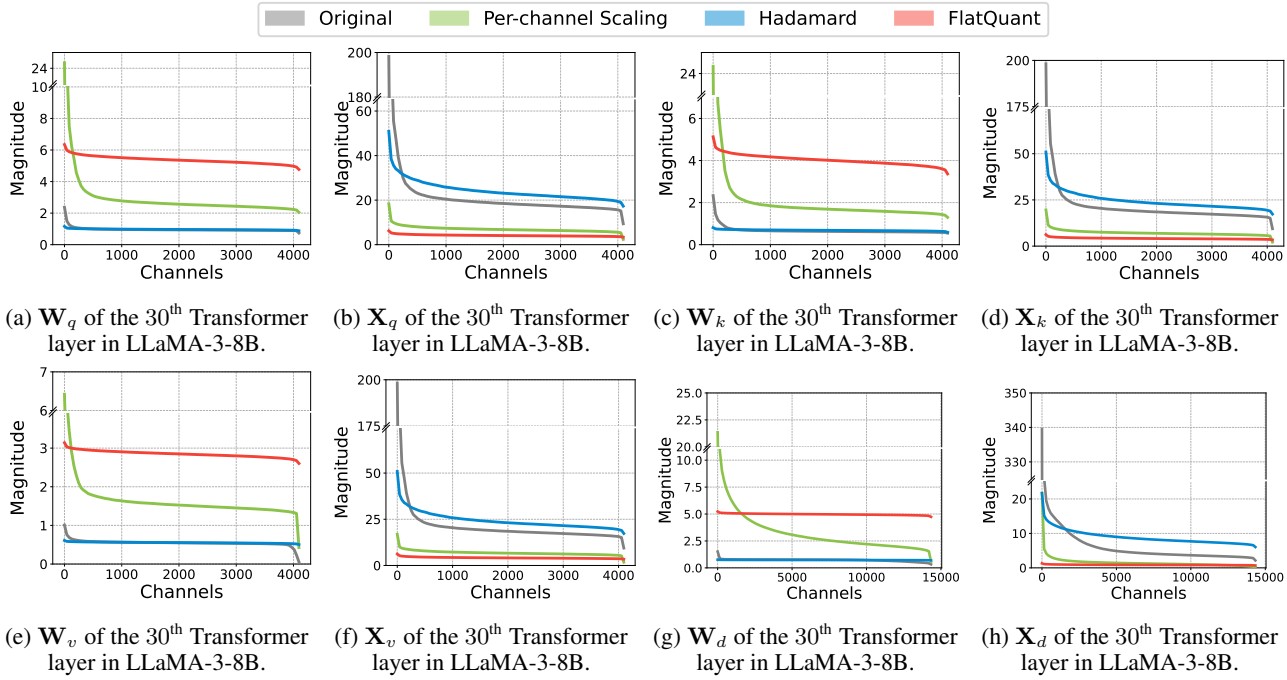

Figure 16: Distributions of weights and inputs from LLaMA-3-8B, sorted by the channel magnitudes (i.e., the Frobenius norm) in descending order.

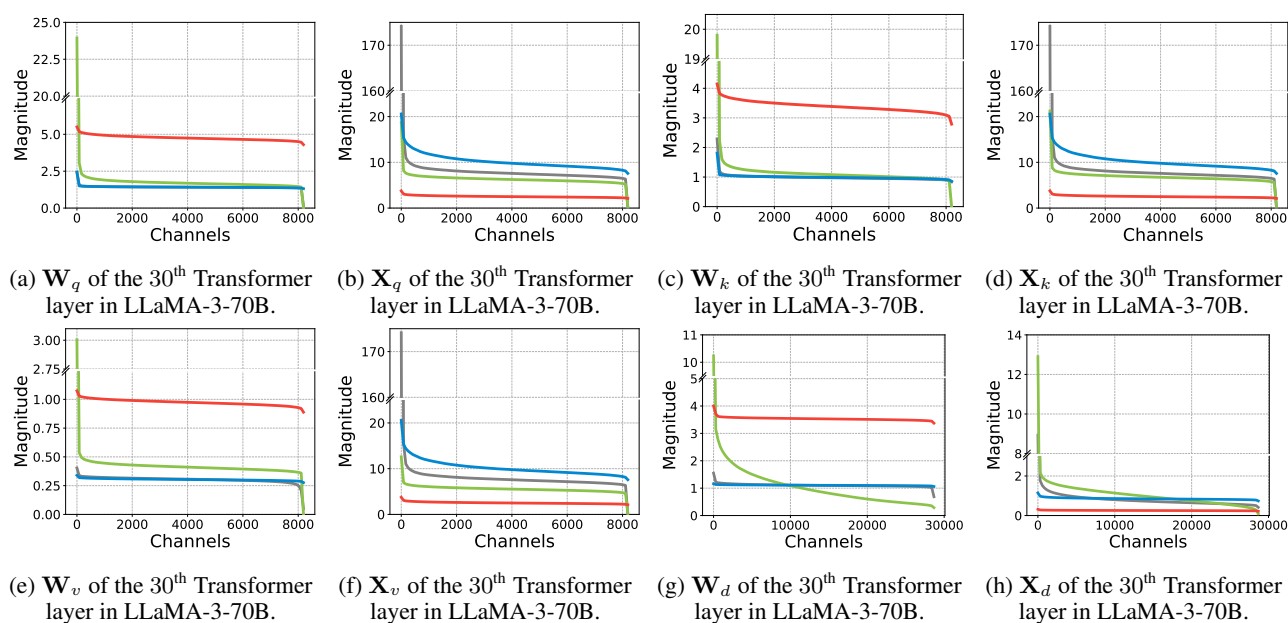

Figure 17: Distributions of weights and inputs from LLaMA-3-70B, sorted by the channel magnitudes (i.e., the Frobenius norm) in descending order.

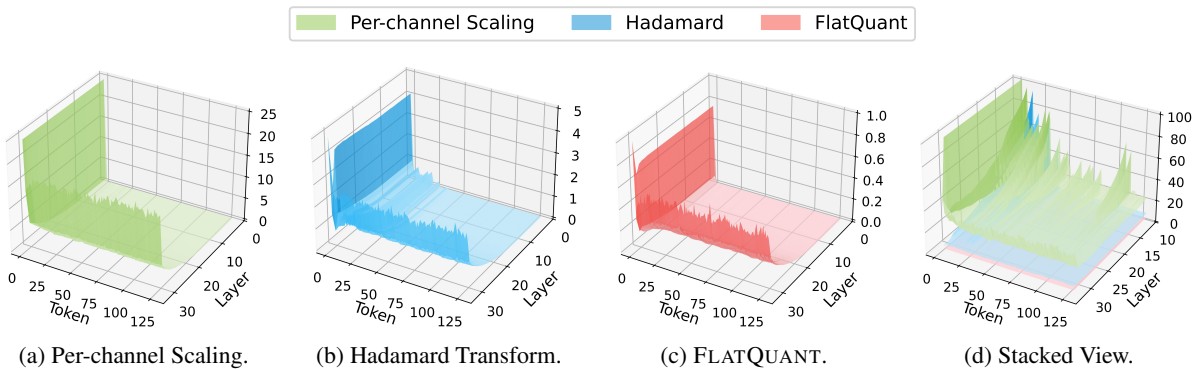

(a) Per-channel Scaling.     (b) Hadamard Transform.     (c) FLATQUANT.     (d) Stacked View.

Figure 18: The MSE of quantization across Transformer layers and input sequence in LLaMA-2-7B. Figure 18a-18c plot the MSE surface of each method, while Figure 18d overlays these surfaces by dividing each MSE with that of FLATQUANT.

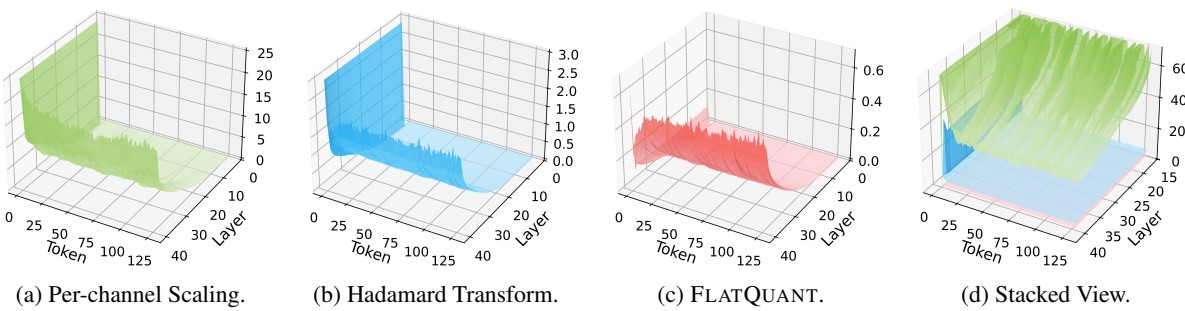

(a) Per-channel Scaling.     (b) Hadamard Transform.     (c) FLATQUANT.     (d) Stacked View.

Figure 19: The MSE of quantization across Transformer layers and input sequence in LLaMA-2-13B. Figure 19a-19c plot the MSE surface of each method, while Figure 19d overlays these surfaces by dividing each MSE with that of FLATQUANT.

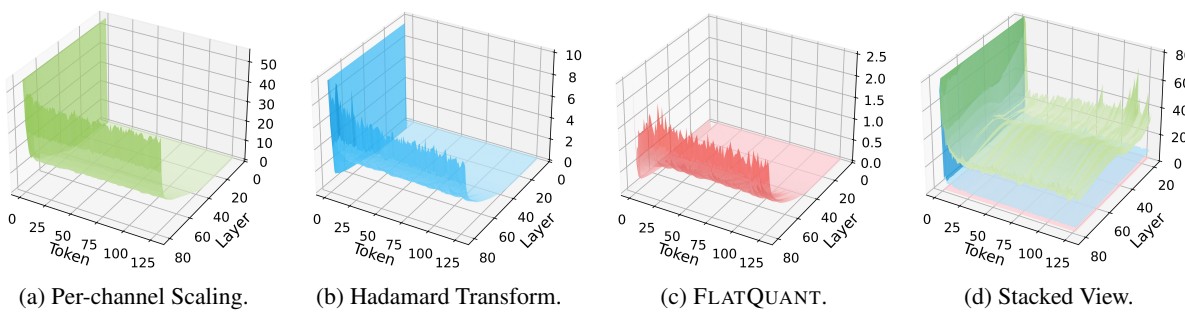

(a) Per-channel Scaling.     (b) Hadamard Transform.     (c) FLATQUANT.     (d) Stacked View.

Figure 20: The MSE of quantization across Transformer layers and input sequence in LLaMA-2-70B. Figure 20a-20c plot the MSE surface of each method, while Figure 20d overlays these surfaces by dividing each MSE with that of FLATQUANT.

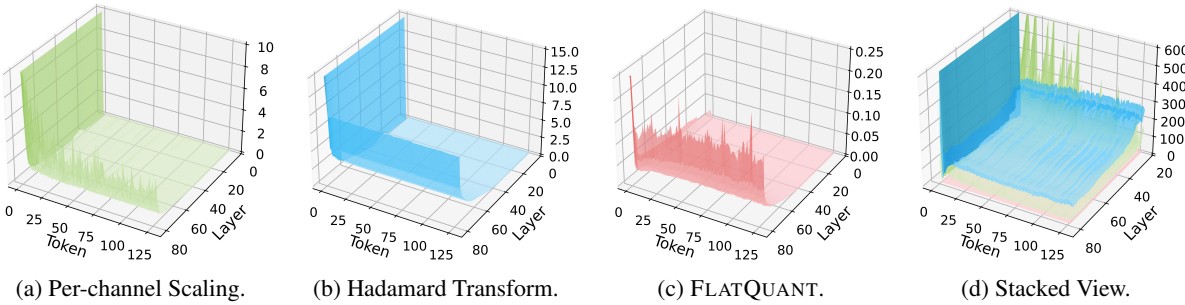

(a) Per-channel Scaling.     (b) Hadamard Transform.     (c) FLATQUANT.     (d) Stacked View.

Figure 21: The MSE of quantization across Transformer layers and input sequence in LLaMA-3-70B. Figure 21a-21c plot the MSE surface of each method, while Figure 21d overlays these surfaces by dividing each MSE with that of FLATQUANT.

