# OpenReview forum: "FlatQuant: Flatness Matters for LLM Quantization"
_ICML.cc/2025/Conference — ICML 2025 poster_

### Official Review · Reviewer_x5Xi · 2025-02-27

**Overall Recommendation:** 4

**Summary:**

This work proposes an approach of learning scaling and rotation transformation to mitigate the problem of outliers and varying dynamic range of weights and activations in LLMs. The learnable rotation transformation is parameterized as a Kronecker product of 2 matrices. Parameters of the transformation are trained to optimize the block reconstruction objective.  The method proposed in validated on several LLM models for W4A4 quantization.

### Update after rebuttal

 I believe than contribution and experimental results are sufficient for acceptance of this work.

**Claims And Evidence:**

The authors claim to achieve new state-of-the-art for W4A4 and the evidence provided does support this statement. Performance drop for W4A4 quantization of Llama-2, Llama-3 models relative to baseline is much smaller than the one produced by baselines (even quite strong, such as QuaRot and SpinQuant).

In addition, FlatQuant comes with certain inference speed-up, and the paper provides an analysis of overhead induces by rotation transformations, making the provided results plausible.

**Essential References Not Discussed:**

Two recent papers with strong performance are not discussed. I would suggest adding them to related work section.

---

References

[1] Lin H. et al. Duquant: Distributing outliers via dual transformation makes stronger quantized llms //Advances in Neural Information Processing Systems. – 2025. – Т. 37. – С. 87766-87800.

[2] Chen, Mengzhao, et al. "Prefixquant: Static quantization beats dynamic through prefixed outliers in llms." arXiv preprint arXiv:2410.05265 (2024).

**Experimental Designs Or Analyses:**

Overall, experimental protocol and choice of baselines is sensible.
However, for an exhaustive comparison I would suggest adding a few more state-of-the-art methods [1, 2]. The former provides other strategy for learning rotation and the latter accounts for outliers tokens in KV cache.

An ablation on impact of each component in the proposed method is provided as well as analysis of computational overhead caused by rotations.

---

References

[1] Lin H. et al. Duquant: Distributing outliers via dual transformation makes stronger quantized llms //Advances in Neural Information Processing Systems. – 2025. – Т. 37. – С. 87766-87800.

[2] Chen, Mengzhao, et al. "Prefixquant: Static quantization beats dynamic through prefixed outliers in llms." arXiv preprint arXiv:2410.05265 (2024).

**Methods And Evaluation Criteria:**

The evaluation protocol adopted is standard for research on LLM compression.

**Other Comments Or Suggestions:**

-

**Other Strengths And Weaknesses:**

-

**Questions For Authors:**

What are the speed-ups on newer generation of GPUs (say GPU 4090) or high-end GPU on prefill and decode? Would you expect them to be lower or higher? In addition, it could be insightful to measure speed-ups with other sequence lengths.

**Relation To Broader Scientific Literature:**

This work can be regarded as a new approach for learning scale and rotation transformation to simplify the problem of quantization.

**Theoretical Claims:**

This work provides primarily a practical contribution. The theoretical motivation of the proposed approach is sound.

---

> ### Author Rebuttal · Authors · 2025-04-01
>
> **Q1**. *For an exhaustive comparison I would suggest adding a few more state-of-the-art methods [1, 2]. The former provides other strategy for learning rotation and the latter accounts for outliers tokens in KV cache.*
>
> *[1] Lin H. et al. Duquant: Distributing outliers via dual transformation makes stronger quantized llms //Advances in Neural Information Processing Systems. – 2025. – Т. 37. – С. 87766-87800.*
>
> *[2] Chen, Mengzhao, et al. "Prefixquant: Static quantization beats dynamic through prefixed outliers in llms." arXiv preprint arXiv:2410.05265 (2024).*
>
>
> **A1**. Thanks for the recommendation regarding the recent works. We will add them in the revised version. PrefixQuant is mainly experimented with static quantization and is a technology orthogonal to our method, we leave the integration of PrefixQuant with FlatQuant as the future work. Below we conduct a detailed comparison between FlatQuant and DuQuant.
>
> * **Comparison with DuQuant in Accuracy.** FlatQuant significantly outperforms DuQuant across multiple benchmarks.
>
> |**LLaMA3-8B**|**WikiText-2**|**C4**|**ARC-C**|**ARC-E**|**HellaSwag**|**LAMBADA**|**PIQA**|**Winogrande**|**Avg.**|
> |:----|:----|:----|:----|:----|:----|:----|:----|:----|:----|
> |FP16|6.14|9.45|53.50|77.57|79.12|75.51|80.74|72.93|73.23|
> |DuQuant|8.13|12.91|44.80|71.30|73.00|68.04|75.73|69.46|67.05|
> |FlatQuant|6.98|11.13|50.00|75.80|76.80|72.91|79.16|72.69|71.23|
>
> * **Comparison with DuQuant in Latency.** DuQuant has the same number of online transformations (each consisting of two matrix multiplications and one channel permutation) as FlatQuant (each consisting of a Kronecker-decomposed matrix multiplication). It can be seen that FlatQuant achieves comparable speedup without kernel fusion and even faster speedup with kernel fusion.
>
> |**Batch Size**|**1**|**4**|**16**|
> |:----|:----|:----|:----|
> |DuQuant|1.95x|2.03x|2.08x|
> |FlatQuant w/o Kernel Fusion|1.94x|2.02x|2.10x|
> |FlatQuant w/ Kernel Fusion|2.12x|2.21x|2.27x|
>
>
> ---
> **Q2**. *What are the speed-ups on newer generation of GPUs (say GPU 4090) or high-end GPU on prefill and decode? Would you expect them to be lower or higher?*
>
>
> **A2**. Thanks for the valuable suggestion. Currently, we haven't implemented INT4 kernels on other GPUs, but we can approximately estimate the speedup. In the following, we estimate the speedup for a single linear layer with hidden dimension $d$ on A100 GPU. The speedup for attention layers can be estimated likewise.
>
> * **A100 Specs**. For A100, we have $\text{Perf} _{\text{BF16}}=312\text{ TFLOPS},\text{Perf} _{\text{INT4}}=1248\text{ TFLOPS},\text{Bandwidth}=1555\text{ GB/s}$
>
> * **Prefill Speedup**. Assume $s=2048$ and $d=4096$ where $s$ is the number of tokens in the prefill. Since in the prefill stage, the linear layer is mostly compute-bound, while the online quantization and transformations are memory-bound, we can estimate the prefill speedup accordingly:
>
> $
> \text{FLOPs} _{\text{Linear}}=2sd^2,
> \text{Mem} _{\text{Quant+Kron}}=2.5sd+4d
> $
>
>
> $
> \text{Prefill Speedup}\approx \frac{\text{FLOPs} _{\text{Linear}}/\text{Perf} _{\text{BF16}}}{\text{FLOPs} _{\text{Linear}}/\text{Perf} _{\text{INT4}}+\text{Mem} _{\text{Quant+Kron}}/\text{Bandwidth}}\approx 3.21
> $
>
> * **Decoding Speedup**. Similarly, the decoding speedup can be estimated with
>
> $
> \text{Mem} _{\text{BF16}}=2d^2+4d,\text{Mem} _{\text{INT4}}=d^2/2+2.5d,\text{Mem} _{\text{Quant+Kron}}=6.5d
> $
>
> $
> \text{Decoding Speedup}\approx \frac{\text{Mem}_ {\text{BF16}}}{\text{Mem}_ {\text{INT4}}+\text{Mem}_ {\text{Quant+Kron}}} \approx 3.98
> $
>
> ---
> **Q3**. *In addition, it could be insightful to measure speed-ups with other sequence lengths.*
>
>
> **A3**. Thanks for the helpful advice. In the following, we provide more speedup results with other sequence lengths as a complement to Figure 4:
>
> * **Prefill** speedup with batch size 1.
>
> |**Prefill Length**|**INT4**|**QuaRot**|**FlatQuant**|
> |:----|:----|:----|:----|
> |2048|2.16x|1.97x|2.12x|
> |4096|2.06x|1.90x|2.04x|
> |8192|1.94x|1.79x|1.92x|
> |16384|1.83x|1.72x|1.80x|
>
> * **Decoding** speedup with batch size 64.
>
> |**KV Cache Length**|**INT4**|**QuaRot**|**FlatQuant**|
> |:----|:----|:----|:----|
> |256|1.38x|1.09x|1.24x|
> |512|1.62x|1.38x|1.56x|
> |1024|1.70x|1.61x|1.63x|
> |2048|1.78x|1.72x|1.76x|

---

> > ### Comment · Reviewer_x5Xi · 2025-04-01
> >
> > Thank you for your response.
> >
> > I believe FlatQuant demonstrates strong performance and significant speed-ups, making it highly promising for practical integration. Therefore, I support the acceptance of this work.

---

> > > ### Author Response · Authors · 2025-04-02
> > >
> > > We greatly appreciate your comments and valuable advice.

---

### Official Review · Reviewer_pwvD · 2025-03-11

**Overall Recommendation:** 4

**Summary:**

This paper presents FLATQUANT, a post-training quantization approach for large language models (LLMs) that focuses on improving the "flatness" of weight and activation distributions to enhance quantization performance. The authors introduce a novel method employing fast and learnable affine transformations tailored to each linear layer in a Transformer model. The approach uses Kronecker decomposition to reduce computational overhead and incorporates kernel fusion for efficient implementation. The paper demonstrates state-of-the-art quantization accuracy with minimal performance degradation, even in challenging W4A4 (4-bit weights and activations) scenarios for large models like LLaMA-3-70B, while providing significant speedups compared to FP16 inference.

**Claims And Evidence:**

The authors make several claims that are generally well-supported by experimental evidence:

Page 1, lines 51-52: The claim that FLATQUANT achieves less than 1% accuracy drop for W4A4 quantization on LLaMA-3-70B is supported by the comprehensive evaluation in Table 2, which shows a minimal drop from 79.95% to 79.01% on average across multiple benchmarks.
Page 2, paragraph 2: The claim that flatter distributions are easier to quantize is well-substantiated through multiple visualizations (Figure 1) and analyses of quantization error landscapes (Figure 2).
Page 3, paragraph 3: The authors claim that Kronecker decomposition reduces memory and computational demands, which is validated by the quantitative analysis showing 2.61% computational overhead and 0.11MB memory overhead (page 4).

**Essential References Not Discussed:**

N/A

**Experimental Designs Or Analyses:**

The experimental analyses are thorough and well-executed:

Page 5, Tables 1-2: The comprehensive evaluation across multiple models (7B to 70B parameters) and datasets provides strong evidence for FLATQUANT's effectiveness.
Page 7, Figure 4-6: The detailed analysis of inference latency demonstrates the practical benefits of FLATQUANT.
Page 8, Figure 7: The correlation between training progress and increasing flatness offers a nice validation of the method's working principle.
Some experimental limitations exist:

Page 7, Figure 5: While the impact of different decomposition sizes is analyzed, this analysis is limited to a single model size (LLaMA-2-7B).
It would be helpful to see more detailed performance profiling across different hardware platforms beyond the RTX3090 GPU mentioned on page 5, line 267.

**Methods And Evaluation Criteria:**

The methods are generally appropriate and well-designed:

The selection of benchmark tasks (WikiText2, C4, and six zero-shot QA tasks) provides good coverage for evaluating language modeling capabilities and task performance.
The comparisons with state-of-the-art baselines like SmoothQuant, OmniQuant, QuaRot, and SpinQuant offer appropriate context for assessing FLATQUANT's contributions.
The ablation studies effectively isolate the contributions of different components (learnable transformations, per-channel scaling, and learnable clipping thresholds).
However, there are methodological limitations:

Page 4, paragraph 3: The calibration procedure using only 128 sentences from WikiText-2 seems limited, though the authors do later show in Table 16 (Appendix) that performance remains consistent across different calibration datasets.
Page 4, line 230-234: The selection of n1 and n2 for Kronecker decomposition is presented as a simple optimization, but the exploration of these parameters is limited to one model size.

**Other Comments Or Suggestions:**

The paper would benefit from a more detailed explanation of why learnable clipping after transformation is more effective than before transformation (as shown in Table 17).
The visualization in Figure 1 effectively illustrates the concept of flatness, but could be improved by adding a metric that quantifies the degree of flatness.
Some minor typos and grammatical issues exist (e.g., "is powerless to such errors" on page 3).

**Other Strengths And Weaknesses:**

Strengths:

Page 3, section 3.1: The Kronecker decomposition approach is a clever solution to the computational overhead problem that plagues prior affine transformation methods.
Page 3, lines 173-179: The training objective for each Transformer block individually simplifies the optimization problem without apparent degradation in performance.
Page 4, section 3.3: The kernel fusion approach is well-designed and practically important for achieving the reported speedups.
Weaknesses:

Page 1, paragraph 1: While the paper claims to establish "new state-of-the-art" results, it would be strengthened by a clearer quantitative comparison of what constitutes the previous SOTA.
Section 2.2: The notion of "flatness" is somewhat intuitive but would benefit from a more rigorous definition or metric.
Page 6, lines 364-368: The claim that "FLATQUANT with RTN is sufficient to be competitive" is intriguing but lacks thorough theoretical explanation.

**Questions For Authors:**

How would FLATQUANT perform when incorporated into more advanced quantization schemes like mixed-precision quantization, where different layers use different bit-widths? This could affect my evaluation as it might demonstrate broader applicability.
The paper mentions that FLATQUANT is robust to initialization (page 4, lines 261-262). Have you analyzed how different initialization strategies for the affine transformation matrices affect convergence speed or final performance?
The paper shows impressive results on LLaMA-3-70B with W4A4 quantization. How would FLATQUANT perform in even more extreme scenarios, such as W3A3 quantization? Some results are shown in Appendix Table 14, but a more detailed analysis of limitations would be valuable.

**Relation To Broader Scientific Literature:**

N/A

**Theoretical Claims:**

The paper's theoretical claims are generally sound and verified:

The idea that flatness matters for quantization is well-established in prior literature, as the authors acknowledge. Their contribution is showing how to better optimize for this objective.
The Kronecker decomposition approach (page 3) is mathematically correct and the efficiency claims are validated by experiments.
I did not identify any issues with the mathematical derivations provided, though the theoretical justifications for why the training process leads to flatter distributions could be more formally developed.

---

> ### Author Rebuttal · Authors · 2025-04-01
>
> **Q1**. ...*a clearer quantitative comparison of what constitutes the previous SOTA.*
>
>
> **A1**. We consider QuaRot and SpinQuant as previous SOTA in Table 1-2, and compare with the latest DuQuant in A1 to Reviewer x5Xi.
>
>
> ---
> **Q2**. *The notion of "flatness" is somewhat intuitive but would benefit from a more rigorous definition or metric.*
>
>
> **A2**. Thanks for the suggestion. We have provided a quantitative definition of flatness in Appendix D.1 Line 1106-1113, and visualized the associated flatness score in Figure 7 and Figure 11.2. We will move this part to the main text.
>
>
> ---
> **Q3**. *The claim that "FLATQUANT with RTN is sufficient to be competitive" is intriguing but lacks thorough theoretical explanation.*
>
>
> **A3**. Thanks for pointing this out. We suspect the affine transformations alone make the distributions friendly to RTN, where the flatness is quantitatively measured in Appendix D.1 and Figure 7. We decide to leave theoretical explanations in future works.
>
>
> ---
> **Q4**. *The paper would benefit from a more detailed explanation of why learnable clipping after transformation is more effective than before transformation (as shown in Table 17).*
>
>
> **A4**. Thanks for the question. This can be attributed to the preservation of important outlier channels.
>
> * **"LCT before Transformation"** consistently performs worse, as it directly clips important outliers in LLMs.
>
> * **"LCT after Transformation"** performs significantly better because the affine transformations redistribute outliers across channels. Even after clipping, the inverse transformation can effectively recover the original outliers.
>
> We will further clarify this issue in our revision.
>
>
> ---
> **Q5**. *Some minor typos and grammatical issues exist.*
>
>
> **A5**. Thanks for the attentive feedback. We will fix these issues in the revised version.
>
>
> ---
> **Q6**. *How would FLATQUANT perform when incorporated into more advanced quantization schemes like mixed-precision quantization, where different layers use different bit-widths? This could affect my evaluation as it might demonstrate broader applicability.*
>
>
> **A6**. Thanks for raising this interesting question. Below we explore some orthogonal mixed-precision strategies to enhance FlatQuant.
>
> * **Weight Importance Score Analysis**. We follow SqueezeLLM[1] to use Fisher information for the estimation of weight importance, and simply sum over the scores to estimate the overall importance of different parts in LLMs, including different linear layer types and Transformer layers.
>
> |**Self-attention**|$\mathbf W_q$|$\mathbf W_k$|$\mathbf W_v$|$\mathbf W_o$|
> |:----|:----|:----|:----|:----|
> |**Score**|44.86|11.01|1139.08|365.17|
>
> |**Feed-forward network**|$\mathbf W_g$|$\mathbf W_u$|$\mathbf W_d$|
> |:----|:----|:----|:----|
> |**Score**|177.20|401.03|2468.97|
>
> |**Top 5 Transformer Layer Index**|**1**|**0**|**2**|**3**|**31**|
> |:----|:----|:----|:----|:----|:----|
> |**Score**|63684.88|18075.19|6740.75|5780.25|5297.59|
>
> * **Mixed-precision Results**. According to the overall importance, we use W8A8 for the top 5 Transformer layers as well as all the down projection layers, which proves to be an effective way to improve FlatQuant.
>
> |**LLaMA-3-8B**|**WikiText-2**|**C4**|**ARC-C**|**ARC-E**|**HellaSwag**|**LAMBADA**|**PIQA**|**Winogrande**|**Avg.**|
> |:----|:----|:----|:----|:----|:----|:----|:----|:----|:----|
> |FP16|6.14|9.45|53.50|77.57|79.12|75.51|80.74|72.93|73.23|
> |FlatQuant|6.98|11.13|50.00|75.80|76.80|72.91|79.16|72.69|71.23|
> |+down_proj_8bits|6.73|10.62|50.00|77.78|77.49|73.96|79.54|70.56|71.55|
> |+Top5_8bits|6.80|10.82|49.23|76.56|76.54|73.51|79.71|73.95|71.58|
> |+Top5&down_proj_8bits|6.61|10.43|51.11|77.74|77.47|74.69|79.92|72.14|72.18|
>
> [1] Kim, Sehoon, et al. "Squeezellm: Dense-and-sparse quantization." arXiv preprint arXiv:2306.07629 (2023).
>
>
> ---
> **Q7**. *Have you analyzed how different initialization strategies for the affine transformation matrices affect convergence speed or final performance?*
>
>
> **A7.** Thanks for the thoughtful feedback. We initialize the transformations with random orthogonal matrices by default and find FlatQuant is generally robust to initialization strategies.
>
> |**Init**|**WikiText-2**|**C4**|**ARC-C**|**ARC-E**|**HellaSwag**|**LAMBADA**|**PIQA**|**Winogrande**|**Avg.**|
> |:----|:----|:----|:----|:----|:----|:----|:----|:----|:----|
> |Identity Matrix|6.98|11.15|49.49|77.10|76.82|74.21|80.09|70.32|71.34|
> |Random Orthogonal Matrix|6.98|11.13|50.00|75.80|76.80|72.91|79.16|72.69|71.23|
>
>
> ---
> **Q8**. *How would FLATQUANT perform in even more extreme scenarios, such as W3A3 quantization? Some results are shown in Appendix Table 14, but a more detailed analysis of limitations would be valuable.*
>
>
> **A8**. Thanks for the advice. As shown in Table 14, although FlatQuant achieves notable improvements over QuaRot, it still suffers significant accuracy degradation. W3A3 needs both improved algorithms and hardware support, which still remains an open question.

---

> > ### Comment · Reviewer_pwvD · 2025-04-03
> >
> > Thanks for the response. I like this work for the research area it explores. I will raise my original rating.

---

> > > ### Author Response · Authors · 2025-04-03
> > >
> > > Thanks for your thorough review and constructive feedback. We sincerely appreciate the time and effort you have dedicated to evaluating our work.

---

### Official Review · Reviewer_7f4H · 2025-03-13

**Overall Recommendation:** 3

**Summary:**

This paper proposes FlatQuant for post-training quantization with learnable linear transformation to remove outliers of weights and activations. The authors propose the use of Kronecker decomposition to reduce the inference overhead. Extensive experiments demonstrate the effectiveness of the proposed approach on W4A4 quantization.

**Claims And Evidence:**

Yes.

**Essential References Not Discussed:**

Regarding techniques for suppressing outliers to enhance quantized LLMs, in addition to per-channel scaling and linear transformations, the authors overlooked the use of infinity-norm regularization for reducing the weight range, e.g., [1,2].

[1] Zhang et al., MagR: Weight Magnitude Reduction for Enhancing Post-Training Quantization.

[2] Kundu et al., R2 Loss: Range Restriction Loss for Model Compression and Quantization.

**Experimental Designs Or Analyses:**

Solid experimental results on LLM quantization.

**Methods And Evaluation Criteria:**

The authors show the effectiveness of the algorithm through extensive experiments.

**Other Comments Or Suggestions:**

N/A

**Other Strengths And Weaknesses:**

Strengths:

1. Proposed Kronecker decomposition to mitigate the computational and memory costs for the additional linear transformation.

2. Demonstrated the effectiveness of the proposed method through extensive experiments.

Weakness:

1. The contributions are somewhat incremental, extending existing work on block-wise quantization and linear transformations for smoothing weights and activation values.

2. The results are primarily empirical, lacking sufficient theoretical justification for the effectiveness of the Kronecker decomposition-based linear transformation. Specifically, it remains unclear why this approach outperforms rotation-based transformations. A deeper theoretical analysis would strengthen the claims.

3. The reported results in Table 1 raise concerns about plausibility. It is counterintuitive that FlatQuant outperforms AffineQuant significantly, given that AffineQuant employs a more general affine transformation, while FlatQuant imposes constraints through low-complexity Kronecker decomposition, trading expressivity for efficiency. Justification is needed to explain why the reduced expressivity yields the better performance.

**Questions For Authors:**

1. What is the role of Kronecker decomposition-based transformation in mitigating outliers? Can we achieve similar effects by imposing alternative low-complexity structures, such as low-rank decomposition?

2. I wonder how FlatQuant performs on 2-bit weight quantization.

**Relation To Broader Scientific Literature:**

N/A

**Theoretical Claims:**

N/A

---

> ### Author Rebuttal · Authors · 2025-04-01
>
> **Q1**. *Regarding techniques for suppressing outliers..., the authors overlooked the use of infinity-norm regularization for reducing the weight range.*
>
>
> **A1**. Thanks for the recommendation regarding related works. We will add the discussions in the revised manuscript.
>
>
> ---
> **Q2**. *The results are primarily empirical, lacking sufficient theoretical justification for the effectiveness of the Kronecker decomposition-based linear transformation. Specifically, it remains unclear why this approach outperforms rotation-based transformations. A deeper theoretical analysis would strengthen the claims.*
>
>
> **A2**. Thanks for the advice. Compared with rotational transformations, FlatQuant relaxes the optimization space from Stiefel manifold to the general linear space, which has more potential to suppress outliers. We leave more detailed theoretical analyses in future work.
>
>
> ---
> **Q3**. *It is counterintuitive that FlatQuant outperforms AffineQuant significantly...*
>
> **A3**. In the following, we discuss the superiority of FlatQuant over AffineQuant in more detail as a complement to Line 631-637:
>
> * **Improved Expressivity**. As discussed in Figure 5, we find that the Kronecker decomposed transformations deliver comparable accuracy with the full-size transformation, which demonstrates its expressivity. In contrast, AffineQuant optimizes strictly diagonally dominant transformation to ensure invertibility, which may restrict its expressivity and make the transformation more like a per-channel scaling (see Figure 7 in the AffineQuant paper).
>
> * **Applicability to All Linear Layers**. FlatQuant applies Kronecker-decomposed linear transformations to all linear layers with minimal overhead. In contrast, AffineQuant directly learns a full-size transformation, and can only apply it to the output projection linear layer for WA quantization so that the transformation can be merged into preceding linear layer to avoid the formidable overhead. The other linear layers can only use per-channel scaling.
>
>
> ---
> **Q4**. *The contributions are somewhat incremental, extending existing work on block-wise quantization and linear transformations for smoothing weights and activation values.*
>
>
> **A4.** We elaborate on the key differences between FlatQuant and previous works below:
>
> * **Linear Transformations v.s. Orthogonal Transformations.** While previous works (e.g. QuaRot) explore orthogonal transformations to smooth outliers, we find linear transformations can achieve better flatness, delivering superior quantization accuracy as shown in Table 1-2.
>
> * **Kronecker Decomposed Matrices with Kernel Fusion.** While linear transformations suffer from high computation and memory consumption, we mitigate it with Kronecker decomposition along with kernel fusion, ensuring practical speedup with minimal overhead.
>
> * **Learnable Clipping After Transformations.** As detailed in Appendix C.4, a key difference of learnable clipping in FlatQuant is that it is applied after the pre-quantization transformations. This can help avoid damaging critical outliers during activation clipping.
>
>
> ---
> **Q5**. *What is the role of Kronecker decomposition-based transformation in mitigating outliers? Can we achieve similar effects by imposing alternative low-complexity structures, such as low-rank decomposition?*
>
>
> **A5**. The learnable transformation to remove outliers needs to satisfy 1) **invertible**, so that the transformation is computational invariance (similar to QuaRot); 2) **lightweight** so that little inference overhead is introduced. Kronecker decomposition is a valid approach to satisfy both conditions, where the invertibility is ensured from (Line 661-663).  However, low-rank decomposition does not satisfy 1) invertibility. We leave the exploration of other alternatives in the future.
>
>
> ---
> **Q6**. *I wonder how FlatQuant performs on 2-bit weight quantization.*
>
>
> **A6**. Thanks for the question. In the following, we experiment with 2-bit asymmetric per-channel weight-only quantization and compare FlatQuant against two strong uniform weight-only quantization methods GPTQ and QuIP[1]. For GPTQ, we use activation reorder and grid search the best weight clipping thresholds.
>
> |**LLaMA-3-8B**|**WikiText-2**|**C4**|**ARC-C**|**ARC-E**|**HellaSwag**|**LAMBADA**|**PIQA**|**Winogrande**|**Avg.**|
> |:----|:----|:----|:----|:----|:----|:----|:----|:----|:----|
> |FP16|6.14|9.45|53.50|77.57|79.12|75.51|80.74|72.93|73.23|
> |GPTQ|**26.76**|247.27|22.70|31.86|35.37|5.20|54.08|51.14|33.39|
> |QuIP|59.96|139.70|23.98|28.24|30.82|1.59|52.18|50.28|31.18|
> |FlatQuant|31.27|**132.02**|**24.49**|**41.04**|**39.42**|**14.28**|**59.03**|**53.75**|**38.67**|
>
> [1] Chee, Jerry, et al. "Quip: 2-bit quantization of large language models with guarantees." Advances in Neural Information Processing Systems 36 (2023): 4396-4429.

---

### Official Review · Reviewer_re4r · 2025-03-14

**Overall Recommendation:** 4

**Summary:**

This paper presents FLATQUANT, a novel post-training quantization framework that enhances the flatness of weight and activation distributions in large language models (LLMs) through learnable affine transformations. FLATQUANT introduces layer-wise affine transformations trained on calibration data to reshape distributions, coupled with Kronecker decomposition to reduce parameter overhead. The method outperforms existing techniques (e.g., QuaRot, SpinQuant) across perplexity, QA accuracy, and inference speed benchmarks, particularly excelling in W4A4 settings. Results on LLaMA-3 models (7B–70B) demonstrate robustness, with kernel fusion further enhancing practical deployment viability.

**Claims And Evidence:**

The authors claim that flatness of weights and activations is crucial for quantization effectiveness, and their method produces flatter distributions than previous approaches. This claim is well-supported by the visualizations in Figure 1, which clearly show the distribution envelopes after different transformations.

The claim that FLATQUANT outperforms previous SOTA methods is convincingly supported by extensive experiments across multiple benchmarks and model scales. For instance, Table 1 and Table 2 show consistent improvements over SpinQuant and QuaRot in perplexity metrics and QA tasks respectively.

**Essential References Not Discussed:**

The authors discuss in depth all relevant work related to the key contributions of this paper.

**Experimental Designs Or Analyses:**

The experimental design is thorough and well-executed. The authors test their method across multiple model sizes (7B to 70B parameters) and across different quantization settings (weight-only, weight-activation, KV cache). The additional experiments in the appendix further bolster confidence in the results.

The kernel fusion experiments (Figure 4 and Table 6) are particularly well-designed, showing how the theoretical speedups translate to practical benefits across different batch sizes and hidden dimensions.

**Methods And Evaluation Criteria:**

The proposed methods align well with the problem at hand. The use of Kronecker decomposition to reduce computational overhead is particularly clever, as it maintains the benefits of affine transformations while minimizing inference costs.

The evaluation criteria are comprehensive, including perplexity on language modeling tasks (WikiText-2, C4) and accuracy on zero-shot QA tasks. The authors also properly evaluate both accuracy and speed metrics, providing a holistic view of the method's practical utility. The comparison against state-of-the-art baselines using the same evaluation framework (lm-eval-harness) ensures fair comparisons.

**Other Comments Or Suggestions:**

1. The paper would benefit from a clearer explanation of how the online transformations are initialized, as initialization strategies can significantly impact convergence and final performance.
2. The hyperlinks throughout the paper appear to be non-functional, which hinders navigation between sections and references. Authors should consider revising their citation formatting to ensure proper link functionality.

**Other Strengths And Weaknesses:**

**Strengths:**

1. The kernel fusion implementation (Section 3.3, Appendix B.3) is particularly impressive, showing how theoretical ideas can be translated into practical performance gains.
2. Figure 2 provides an insightful visualization of quantization error propagation across layers and tokens, helping explain why flatter distributions matter.
3. The method's effectiveness across different model architectures (shown in the supplementary material) speaks to its generalizability.

**Weaknesses:**
1. The authors do not provide a comparative assessment of GPU memory consumption during inference relative to baseline methods. This omission is particularly significant for edge deployment scenarios where memory constraints are often the primary limiting factor.
2. The paper lacks exploration of potential biases introduced by Kronecker decomposition and how these biases might affect quantization error propagation through the network. Understanding this relationship would strengthen the theoretical foundation of the approach.
3. While FLATQUANT incorporates LCT and shows improvements over SpinQuant, the paper doesn't adequately isolate whether this specific component is the primary driver of performance gains. A more targeted ablation would clarify the relative importance of affine transformations versus clipping thresholds.
4. The authors treat each layer's quantization independently, but don't thoroughly examine how quantization decisions in earlier layers affect the optimal transformations for later layers, potentially overlooking cascading effects.

**Questions For Authors:**

Please refer to Strengths And Weaknesses.

**Relation To Broader Scientific Literature:**

The authors provide a comprehensive discussion of related work, clearly positioning FLATQUANT relative to existing methods like SmoothQuant, OmniQuant, AffineQuant, QuaRot, and SpinQuant. This paper builds upon the insight from previous work that outliers in LLMs pose challenges for quantization, but extends beyond prior approaches by learning model-specific transformations rather than relying on fixed transformations like Hadamard.

**Theoretical Claims:**

The paper makes minimal purely theoretical claims, focusing instead on empirical findings. The explanation of how Kronecker decomposition reduces computational complexity (line 205-213) appears sound, showing that memory requirements are reduced by a factor of approximately n/2 when the dimensions are balanced (n1 ≈ n2 ≈ √n)..

---

> ### Author Rebuttal · Authors · 2025-04-01
>
> **Q1**. *The authors do not provide a comparative assessment of GPU memory consumption...*
>
>
> **A1**. Thanks for the helpful advice. Below, we provide more results on the memory consumption. The experiment results are consistent with our theoretical analysis in Appendix B.2 Line 736-739, demonstrating the minimal memory overhead brought by the online transformations in FlatQuant.
>
> * **Peak memory usage for decoding a single token** on one Transformation layer of LLaMA-2-7B model with KV caches of different lengths and batch size 1.
>
> |**Sequence Length**|**FP16 (GB)**|**INT4 (GB)**|**FlatQuant (GB)**|**Saving Factor**|
> |:----|:----|:----|:----|:----|
> |256|0.393|0.110|0.110|3.58x|
> |512|0.399|0.112|0.112|3.56x|
> |1024|0.411|0.118|0.118|3.48x|
> |2048|0.434|0.130|0.130|3.35x|
>
>
> ---
> **Q2**. *The paper lacks exploration of potential biases introduced by Kronecker decomposition...*
>
>
> **A2**. Thanks for the thoughtful feedback. As shown in Figure 5, experiments demonstrate that Kronecker decomposition preserves expressivity, achieving accuracy comparable to full-size transformations. We leave theoretical justifications as future work.
>
>
> ---
> **Q3**. ...*A more targeted ablation would clarify the relative importance of affine transformations versus clipping thresholds.*
>
>
> **A3**. Thanks for the constructive advice. Below we provide more results and discussions to demonstrate the effectiveness of LT:
>
> * **Full Ablation Table**. We provide full ablation results as a complement to Table 3, which clearly showcases the effectiveness of each part in FlatQuant, especially LT.
>
> |**LT**|**PS**|**LCT**|**WikiText-2**|**C4**|**ARC-C**|**ARC-E**|**HellaSwag**|**LAMBADA**|**PIQA**|**Winogrande**|**Avg.**|
> |:----|:----|:----|:----|:----|:----|:----|:----|:----|:----|:----|:----|
> ||||1266.60|936.41|25.26|28.62|27.04|1.26|51.80|51.93|30.99|
> ||√||NaN|NaN|22.70|25.08|25.04|0.00|49.51|49.57|28.65|
> |||√|1149.08|1490.08|22.95|29.29|27.35|0.60|52.99|50.83|30.67|
> ||√|√|8197.96|4654.07|25.43|25.72|25.96|0.02|50.49|48.86|29.41|
> |√|||8.50|13.51|44.97|71.38|73.17|67.05|76.88|67.48|66.82|
> |√|√||7.95|12.74|44.20|71.89|74.21|68.72|77.15|66.30|67.08|
> |√||√|7.11|11.47|49.32|76.14|76.30|72.17|78.89|71.51|70.72|
> |√|√|√|6.98|11.13|50.00|75.80|76.80|72.91|79.16|72.69|71.23|
>
> * **Comparison between FlatQuant's affine transformation and SpinQuant's orthogonal transformation**. For both methods, we use asymmetric activation quantization following SpinQuant and do not activate weight or activation clipping. For FlatQuant, we also do not use PS. It can be seen that the affine transformations in FlatQuant are more effective in easing outliers, leading to higher accuracy especially when RTN is used.
>
> |**LLaMA3-8B**|**WikiText-2**|**C4**|**ARC-C**|**ARC-E**|**HellaSwag**|**LAMBADA**|**PIQA**|**Winogrande**|**Avg.**|
> |:----|:----|:----|:----|:----|:----|:----|:----|:----|:----|
> |FP16|6.14|9.45|53.50|77.57|79.12|75.51|80.74|72.93|73.23|
> |SpinQuant-RTN|41.15|63.89|24.15|38.09|45.12|20.80|59.47|55.01|40.44|
> |FlatQuant-LT-RTN|**8.05**|**12.85**|46.08|71.30|73.74|67.15|76.28|68.59|**67.19**|
> |SpinQuant-GPTQ|7.95|13.44|44.80|73.36|73.79|66.99|76.01|68.11|67.18|
> |FlatQuant-LT-GPTQ|**7.66**|**12.58**|46.16|73.57|74.83|67.55|76.01|67.32|**67.57**|
>
>
> ---
> **Q4**. *The authors treat each layer's quantization independently, … , potentially overlooking cascading effects.*
>
>
> **A4**. Thanks for this valuable question. We tried to minimize the MSE loss between ground truth and outputs given quantized inputs instead of given full-precision inputs (as in Equation 4) to mitigate cascading effects. However, results show that full-precision inputs lead to better performance. To learn the mechanism behind, we think this is similar to the "teacher forcing" in training RNNs (i.e., use the ground truth instead of auto-regressive output for training at each step). Similar effects were also verified to be helpful in model quantization [1].
>
> |**LLaMA-3-8B**|**WikiText-2**|**C4**|**ARC-C**|**ARC-E**|**HellaSwag**|**LAMBADA**|**PIQA**|**Winogrande**|**Avg.**|
> |:----|:----|:----|:----|:----|:----|:----|:----|:----|:----|
> |FP16|6.14|9.45|53.50|77.57|79.12|75.51|80.74|72.93|73.23|
> |FlatQuant|6.98|11.13|50.00|75.80|76.80|72.91|79.16|72.69|71.23|
> |+quant_inps|7.14|11.51|50.68|76.64|75.77|70.25|80.41|68.51|70.38|
>
> [1] Bai, Haoli, et al. "Towards efficient post-training quantization of pre-trained language models." Advances in neural information processing systems 35 (2022): 1405-1418.
>
>
> ---
> **Q5**. *The paper would benefit from a clearer explanation of how the online transformations are initialized,....*
>
> **A5**. Please see A7 to Reviewer pwvD.
>
>
> ---
> **Q6**. *The hyperlinks throughout the paper appear to be non-functional,...*
>
>
> **A6**. We have double-checked the manuscript and the hyperlinks function well on our side. We are sorry for the inconvenience, and maybe a different PDF browser could resolve your issue.

---

> > ### Comment · Reviewer_re4r · 2025-04-04
> >
> > This paper is well-experimented and theoretically sound. Therefore, I am inclined to accept this paper. I will raise my score from 3 to 4.

---

> > > ### Author Response · Authors · 2025-04-07
> > >
> > > Thanks for your valuable comments. We are sincerely grateful for your thoughtful review process.

---

### Decision · Program_Chairs · 2025-05-01

**Decision:**

Accept (poster)

**Comment:**

This paper presents FlatQuant, a novel post-training quantization method for large language models (LLMs) that effectively enhances the flatness of weight and activation distributions through learnable affine transformations combined with Kronecker decomposition. Reviewers found the paper's contributions impactful, highlighting strong empirical results, comprehensive evaluations across model scales, and practical inference speedups as notable strengths. I also think the kernel fusion results showing the practical speedup is a nice contribution.

Reviewers unanimously recommended acceptance after the rebuttal, appreciating the extensive experimental validation, clarity of presentation, and innovative approach to managing outliers in quantization.

However, several suggestions remain for improving the final manuscript:

- Clearly present memory usage comparisons in the main text.

- Include detailed ablation results from reviewer re4r to further illustrate the role and effectiveness of the learnable transformations.

- Further clarify in the main text why FlatQuant outperforms AffineQuant.

- Clearly integrate and discuss mixed-precision quantization findings.

- Include comparison with DuQuant within the main experimental results section.

Overall, the paper significantly advances the state-of-the-art in LLM quantization and offers practical benefits with demonstrable inference improvements. Based on the reviewers' feedback and the thorough responses provided by the authors, my recommendation is to accept this submission.